# Scaling Synthetic Task Generation for Agents via Exploration

**Ram Ramrakhya, Andrew Szot, Omar Attia, Bogdan Mazoure, Anh Nguyen,
Yuhao Yang, Zhe Gan, Harsh Agrawal, Alexander Toshev**

## Abstract

Post-Training Multimodal Large Language Models (MLLMs) to build interactive agents holds promise across domains such as computer-use, web navigation, and robotics. A key challenge in scaling such post-training is lack of high-quality downstream agentic task datasets with tasks that are diverse, feasible, and verifiable. Existing approaches for task generation rely heavily on human annotation or prompting MLLM with limited downstream environment information, which is either costly or poorly scalable as it yield tasks with limited coverage. To remedy this, we present AUTOPLAY, a scalable pipeline for task generation that explicitly explores interactive environments to discover *possible interactions* and *current state* information to synthesize environment-grounded tasks. AUTOPLAY operates in two stages: (i) an exploration phase, where an MLLM explorer agent systematically uncovers novel environment states and functionalities, and (ii) a task generation phase, where a task generator leverages exploration trajectories and a set of task guideline prompts as context to synthesize diverse, executable, and verifiable tasks. We show AUTOPLAY generates 20k tasks across 20 Android applications and 10k tasks across 13 applications Ubuntu applications to train mobile-use and computer-use agents. AUTOPLAY generated tasks enable large-scale task demonstration synthesis without human annotation by employing an MLLM task executor and verifier. This data enables training MLLM-based UI agents that improve success rates up to $20.0\%$ on mobile-use and $10.9\%$ on computer-use scenarios. In addition, AUTOPLAY generated tasks combined with MLLM verifier-based rewards enable scaling reinforcement learning training of UI agents, leading to an additional $5.7\%$ gain. coverage. These results establish AUTOPLAY as a scalable approach for post-training capable MLLM agents reducing reliance on human annotation.

## 1 Introduction

Multimodal Large Language Models (MLLMs) are a promising foundation for building agents across a wide range of downstream domains, including computer use (Qin et al., 2025; Agashe et al., 2025; OpenAI, 2025), web navigation (Zhou et al., 2023; Yao et al., 2024; 2022), video games (Fan et al., 2022; Wang et al., 2023), and robotics (Driess et al., 2023; Black et al., 2024; Kim et al., 2024). Owing to their broad knowledge and reasoning capabilities, these models are well-suited to understanding and planning the execution of open-ended tasks across these diverse application areas. A central challenge in training such agents is the scarcity of downstream interactive agentic data. Such interactive data consists of two components: (1) Diverse Tasks: a sufficiently broad set of tasks or queries that cover real-world use cases of such agents, (2) Task demonstrations: corresponding task execution trajectory. For most of these domains, such data is not readily available at web-scale, as much of the relevant interaction data resides on closed systems such as personal devices and commercial hardware. Consequently, post-training efforts for building agentic MLLMs for such domains have relied heavily on human annotation to source a large pool of diverse tasks and corresponding demonstrations (Li et al., 2024; Wang et al., 2025; Qin et al., 2025). However, this approach is prohibitively expensive and scales poorly. We argue, the fundamental bottleneck to enable scalable post-training of agentic MLLMs is lack of large high-quality task definition datasets with tasks that are diverse, feasible to execute, verifiable, and aligned with real-world use cases. With access to such task datasets, MLLMs could be post-trained in a scalable manner with synthetically generated demonstrations using supervised finetuning (SFT) or via Reinforcement Learning (RL)

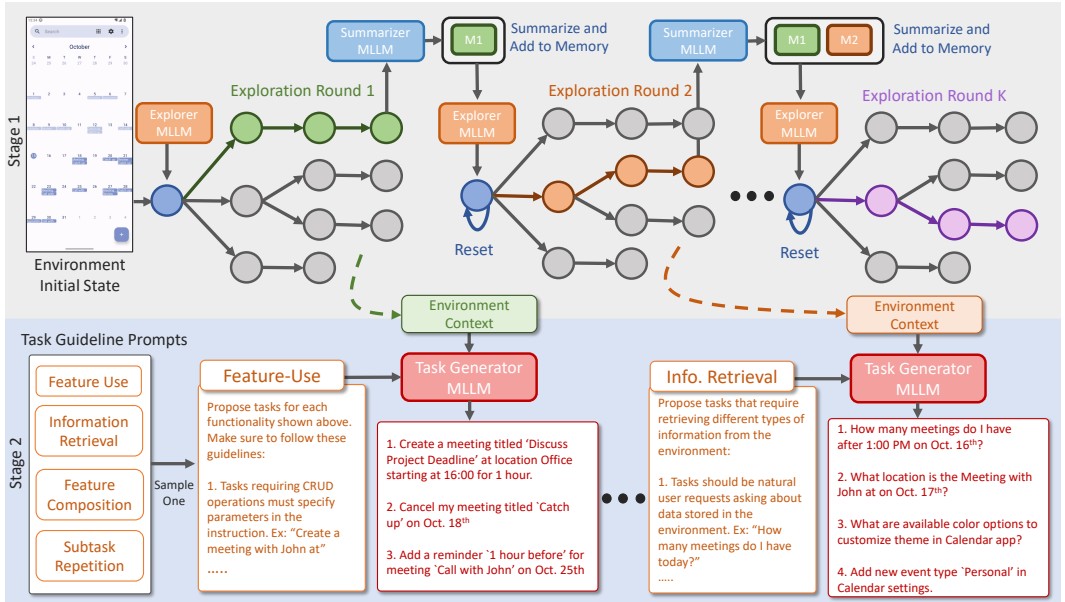

**Figure 1.** AUTOPLAY generates large-scale, diverse and verifiable tasks for scaling supervision for MLLM agents. In stage 1 (top), AUTOPLAY covers the environment states through a MLLM exploration policy that tracks seen states via a memory module. Next, stage 2 (bottom) uses these exploratory trajectories and task guideline prompts as context for proposing tasks. The guidelines help enforce task diversity and the exploration trajectories uncover environment features and content relevant for proposing tasks.

without any human annotation. This highlights the need for an automatic and reliable pipeline to synthesize tasks at scale.

For synthesized task datasets to be useful, they must provide broad *coverage* of the target environment, ensuring a diverse and representative training set. They must also be *feasible*, meaning that agents can realistically execute them to generate useful demonstrations. Finally, they should be *verifiable* to filter high-quality trajectories for SFT or to build a reward model for RL. However, building an automatic pipeline that generates tasks with these properties is challenging as it requires explicit knowledge of current state information and what interactions are feasible in an environment – knowledge that can only be obtained through direct interaction with the environment.

In an attempt to address this challenge, prior works rely on limited domain knowledge of an MLLM to generate the tasks (Trabucco et al., 2025; Zhou et al., 2025), this typically yields generic tasks with limited coverage and offers no guarantees of feasibility or verifiability as current MLLMs lack explicit grounding in both environment dynamics and current state information. To address these shortcomings, we propose an approach that actively explores the agent's environment in an exhaustive manner, collecting relevant information that can be used to ground MLLMs in the environment for scalable task generation. In this work, we focus on UI agents (Rawles et al., 2024; Xie et al., 2025) that observe the current screenshot and interact with UI elements as a human user would. This setting is both broad, capturing the full spectrum of everyday device tasks and practical since there are programmatic environments available for UI interactions (Rawles et al., 2024; Xie et al., 2024).

We introduce AUTOPLAY an approach for scalable task generation with a focus on task coverage, feasibility, and verifiability. Our method, depicted in Figure 1, incorporates two phases of exploration and task generation. First, in *environment exploration* phase, an MLLM explorer agent equipped with memory is prompted to exhaustively explore an increasing number of novel environment states (top of Fig. 1). Such exploratory trajectories are intended to discover the accessible functionalities and content of the environment. Next, in *task generation* phase, a task generator MLLM uses exploration trajectories as environment context to produce diverse environment-grounded tasks based on a set of task guideline prompts which describe desired task properties (bottom of Fig. 1). For instance, a task guideline prompt for Feature-Use tasks would encourage generation of tasks that require doing diverse create, edit, of delete operations on entities in the environment. We present an example of the exploration trajectory collected by AUTOPLAY and the corresponding tasks synthesized using task generator grounded in the state of the environment in Fig. 2.

We use AUTOPLAY to scale task generation for mobile and computer UI agents. AUTOPLAY generates 20k tasks across 20 apps in an Android platform and 10k tasks across 13 apps in an Ubuntu platform. We then synthesize demonstrations for the AUTOPLAY-generated tasks using an MLLM executor and verify them with an MLLM verifier, without relying on privileged environment information or human annotation. These demonstrations are used to finetune an MLLM agent with SFT. Additionally, AUTOPLAY generated tasks also enable training the MLLM agent with RL, using the MLLM verifier as a reward model. Through these experiments we demonstrate AUTOPLAY enables post-training MLLM agents using both SFT and RL in a scalable manner without relying on any human annotation.

We show that the AUTOPLAY pipeline is able to train effective MLLM UI agents. In mobile-use, AUTOPLAY boosts success rate performance by 13%–20% over the base model across a range of model sizes. In computer-use, our method improves base model success rate performance by up to 10.9%. Beyond SFT, incorporating the generated tasks with the verifier for RL training yields an additional 5.7% improvement in task success rate. Together, these results demonstrate that AUTOPLAY-generated tasks and trajectories lead to consistent and significant performance gains across model sizes and training paradigms. We also find that the AUTOPLAY task generation pipeline significantly outperforms prior approaches for synthetically generating tasks in the mobile-use domain (Trabucco et al., 2025; Zhou et al., 2025; Pahuja et al., 2025; Xie et al., 2025) which involve either generating tasks with limited environment information or methods that interleave task proposal with execution. We find this improvement is a result of AUTOPLAY generating tasks with higher diversity, coverage, and feasibility for task execution than prior methods, ultimately yielding datasets that better support the training of capable UI agents.

## 2  RELATED WORK

**Zero-Shot Methods for Agentic Tasks**. Modular agentic pipelines (Agashe et al., 2025; Koh et al., 2024) that leverage MLLMs as high-level task planners, combined with diverse tools (*e.g.* low-level action controllers, memory, and tool use), have emerged as an effective approach for building capable UI agents across various domains (Xie et al., 2024; Rawles et al., 2024; Yao et al., 2024; Deng et al., 2023; Zhou et al., 2023). These pipelines aim to decompose the skills required for complex tasks–such as mapping high-level actions to environment-specific low-level controls, maintaining interaction history, verifying execution outcomes, and invoking tools–into specialized modules. Such modules can then be flexibly composed, enabling scalable and adaptable modular agentic pipelines. In this work, we leverage these agentic pipelines as synthetic data generation modules that enable us to collect UI-interaction demonstrations for AUTOPLAY generated tasks to bootstrap training end-to-end UI agents that take environment observation as input and directly output actions.

**Synthetic Task and Trajectory Generation**. Synthetic data generation using agentic pipelines has emerged as a promising approach to unlock internet-scale data for training UI agents. For instance, methods like PAE Zhou et al. (2025) leverage LLMs conditioned on limited information about the environment (*i.e.* manually written textual descriptions) to propose tasks to synthesize task execution trajectories. In contrast, AUTOPLAY explicitly explores the environment to gather richer context about the environment without relying on human annotated textual descriptions. Another line of research focuses on iteratively proposing and executing tasks. Xie et al. (2025); Pahuja et al. (2025) use an initial screenshot from the environment and propose a short-horizon subtask, and then execute this subtask, repeating this process and using the summary of previously executed sub-task as context to propose the next subtask. This chain of subtasks is then summarized into a single long-horizon task with hindsight relabeling. Similarly, Trabucco et al. (2025) iteratively proposes and executes subtasks, but only for a single iteration without chaining subtasks. Like AUTOPLAY these methods ground tasks in environment interactions, yet they are limited to each trajectory directly mapping to an instruction via hindsight relabeling. AUTOPLAY can propose multiple tasks from a single trajectory based on the uncovered possible interactions and sampled task guideline prompt. Furthermore, these prior works require each iterative subtask to start from the previous subtask, which progressively constrains the space of exploratory trajectories. On the other hand, methods like (Murty et al., 2024; 2025) start by using an LLM as a unconstrained exploration policy, next a trajectory labeler summarizes the exploration trajectories at regular intervals to a task instruction. Next, a verifier verifies whether the trajectory completes the proposed task instruction or not. If it does not, a instruction following policy attempts to execute the proposed task instruction. In contrast to our method, (Murty et al.,

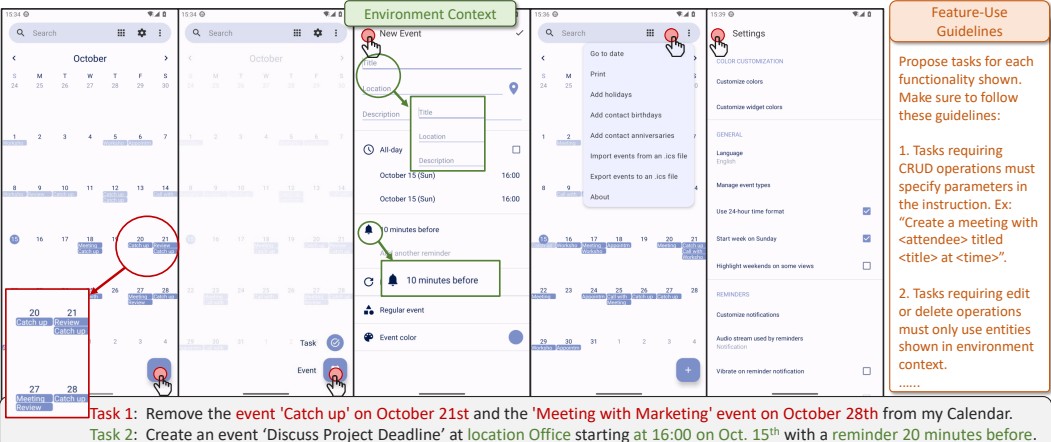

**Figure 2.** Example of task generation based on an environment context, represented as a set of screenshots and interactions, and a set of task guidelines. AUTOPLAY uses presences of events in the calendar together with guidance of using entities in the context, such as names and dates, to produce Task 1. Similarly, showing the event creation form in the context, coupled with guidance to use these fields as task parameters, results in Task 2.

2024; 2025) does not use an exploration policy with memory and explicit task guidelines to maximize coverage of the environment or maximizing task diversity.

## 3 AUTOPLAY

### 3.1 PRELIMINARIES

We define AUTOPLAY in the context of multi-step decision making domains that be expressed as Partially Observable Markov Decision Processes (POMDP) (Puterman, 2014). A POMDP can be defined as a tuple $(\mathcal{S}, \mathcal{O}, \mathcal{A}, P, R, \rho_0)$ for underlying state space $\mathcal{S}$, observation space $\mathcal{O}$, action space $\mathcal{A}$, transition function $P : \mathcal{S} \times \mathcal{A} \to \Delta(\mathcal{S})$, reward function $R$ and initial state distribution $\rho_0$. We consider the extension of including a goal distribution $\mathcal{G}$ and the case where the reward is formulated as $R(s, g)$ for $s \in \mathcal{S}$ and $g \in \mathcal{G}$. We seek to learn a goal-conditioned policy $\pi(a_t|o_t, g)$ mapping from observation $o_t$ at timestep $t$ and goal $g$ to an action $a_t$ to achieve the goal state $s_g$. We train a goal-conditioned policy $\pi(a|o, g)$ in a POMDP with a dataset of tasks $\mathcal{D}$. Using the above formalism, dataset of task $\mathcal{D}$ consists of tuples $(g, s_0)$ where $g$ is a goal-specification in natural language, and $s_0$ is the initial state of the environment. Similarly, we define trajectory as a sequence $\tau = (o_0, a_0, \ldots, o_T)$ where the outcome of achieving goal $g$ is verified by a reward model. In our case, the reward model only relies on the trajectory $R(\tau, g)$ without privileged access to the environment state.

The goal of AUTOPLAY is to automatically generate large dataset of tasks $\mathcal{D}$ for a specific domain by actively interacting with POMDP to gather state information, semantics, and understanding dynamics. With access to task dataset $\mathcal{D}$ one can either attempt to generate training trajectories using a data collection policy for supervised finetuning (SFT), or use the reward model $R$ in a multi-task reinforcement learning (RL) setup to train a goal-conditioned policy $\pi(a|o, g)$ . We instantiate AUTOPLAY in the UI agent domain for mobile-use and computer-use environments, e.g. a Calendar app on a mobile/desktop device. In this domain a state observation $\mathcal{O}$ is the partially observable device screenshot, an action is a UI interaction such as 'click', 'type', 'scroll', etc., and the transition operator is defined by transition dynamics of the UI application. A task is defined by a goal like $g =$"Remove all the events on my calendar for next Tuesday" and initial state $s_0$ starts with the agent on the homepage of the application with data populated in the Calendar application.

### 3.2 AUTOPLAY TASK GENERATOR

AUTOPLAY task generator operates in two stages: First, the POMDP is explored in a goal-agnostic manner to maximize coverage over novel states of the environment using an explorer agent. Second, the task dataset $\mathcal{D}$ is produced using the information of the explored states combined with domain specific task guidelines.

---

**Algorithm 1** AUTOPLAY

| | |
|---|---|
| **Stage 1: Environment Exploration** | **Stage 2: Task Generation** |

**Stage 1: Environment Exploration**

> **Parameters:**
> $N$ # of apps
> $M$: # of exploration turns
> $\mathcal{E} = \emptyset$
> **for** $j = 1 \ldots N$ **do**
>> Sample $s_0 \sim \mathcal{S}$, Initialize context $M = \emptyset$
>> **for** $k = 1 \ldots M$ **do**
>>> Sample trajectory $\tau$ using MLLM($M$, explorer_prompt)
>>> Summarize as $m = \text{MLLM}(\tau, \text{summary\_prompt})$
>>> $\mathcal{E} = \mathcal{E} \cup \{\tau\}$, $M = M \cup \{m\}$
>> **end for**
> **end for**

**Stage 2: Task Generation**

> **Parameters:**
> $\mathcal{P}$: task guidelines
> $K$ # of tasks per guideline and context
> $\mathcal{D} = \emptyset$
> **for** $s = 1 \ldots S$ **do**
>> Sample $p \sim \mathcal{P}$, $\tau \sim \mathcal{E}$
>> Sample $(g_1, \ldots, g_K)$ using
>>> MLLM(task_generator_prompt, $p$, $\tau$, )
>> $\mathcal{D} = \mathcal{D} \cup \{(g_1, s_1), \ldots, (g_K, s_1)\}$
>>> where $s_1$ is the first state in $\tau$
> **end for**
> **return** $\mathcal{D}$

---

**Stage 1: Environment Exploration**. In this stage, a goal-agnostic explorer agent is employed to exhaustively explore the environment to maximize coverage over novel states and functionalities of the environment. The explorer agent is implemented using an MLLM agent equipped with an explicit memory of past interactions. At each timestep, the MLLM agent is provided current environment observation, past interaction memory, and prompted to select diverse actions, aiming to cover the full range of possible interactions within an environment. The explorer agent interacts with the environment for $K$ steps. This process produces a exploration trajectory, denoted as $\tau = (o_1, a_1, \cdots, o_K, a_K)$, which we refer to as the *environment context*.

To ensure the environment is explored as exhaustively as possible, we repeat the exploration process for each environment $M$ times. To encourage the explorer to cover novel states we provide explorer access to prior exploration turns $\{\tau_1, \cdots, \tau_{i-1}\}$ *i.e. episodic memory*. As MLLMs have finite context window, directly conditioning on entire high-dimensional past trajectories is infeasible. Therefore, for each exploration turn $i$, we summarize the prior exploration turns $\{\tau_1, \cdots, \tau_{i-1}\}$ to $\{m_1, \cdots, m_{i-1}\}$ where $m$ is a concise and comprehensive text representation describing exploration trajectory $\tau$ generated using a summarizer MLLM by $m = \text{MLLM}(\text{summary\_prompt}, \tau)$. At the end of stage 1, we obtain a full *environment context* $\mathcal{E} = \{\tau_1, \cdots, \tau_M\}$.

**Stage 2: Exploration Conditioned Task Generation** In this stage, each environment context from $\tau \sim \mathcal{E}$ is used to define a set of $N$ tasks. We would like to highlight that, the environment context $\tau$ does not represent a successful execution of a task, rather it serves as context of what interactions are feasible and current state of the environment which can be used as hints to synthesize tasks. More concretely, the environment context $\tau$ helps uncover underlying information of the POMDP transition function $\mathcal{T}$ and state space $\mathcal{S}$. We use a MLLM as a task generator that uses $\tau$ to generate environment-grounded tasks. Since there are many possible ways to derive task instructions from a single environment context, we find it useful to provide domain-specific prompts that specify guidelines for what constitutes a good task to our task generator MLLM. These prompts, $\mathcal{P} = \{p_1, \ldots, p_N\}$, referred to as *task guidelines*, are tailored to the target domain. For the UI domain, they are illustrated in Fig. 2. For example, one guideline, called Feature-Use, encourages creation of tasks that require doing diverse create, edit, of delete operations on entities in the environment shown in the corresponding environment context.

To produce the task dataset $\mathcal{D}$, we append the task generator prompt, with trajectory data and task guideline prompt. We iterate over all combinations of trajectories in environment context $\tau \in \mathcal{E}$ and task guideline prompt $p \in \mathcal{P}$ to sample a sequence of goals $(g_1, \cdots, g_N) \sim \text{MLLM}(\text{task\_generator\_prompt}, p, \tau)$. Each goal, when paired with the initial state of $\tau$, results in the task dataset. An example of this process for the UI domain is shown in Fig 2.

The above approach is summarized in Algorithm 1. The prompts used for the explorer, summarizer, and task generator MLLMs are described in Appendix C.2. The explorer agent follows the same architecture as the task executor agent detailed in Section 3.3.

### 3.3 TRAINING DATA GENERATION AND VERIFICATION

In the following, we describe the task executor and verifier used in the UI domain.

**Task Executor Agent**. We use the tasks generated by AUTOPLAY to produce successful trajectories via an MLLM based executor agent. Our executor agent builds on the system proposed in Yang

et al. (2025) and consists of an MLLM planner, a reflection tool, and a grounding model. Given the task instruction, the current screenshot, previously executed actions, and a reflection trace of the last action generated by reflection tool, the MLLM planner generates a high-level action in natural language. If this high-level action is coordinate-based (e.g., a click), the grounding model translates it into a pixel-level coordinate for execution. Non-coordinate-based actions are executed directly. The reflection tool supplements this process by taking as input the previous action, the prior observation, and the current observation to generate a reflection trace. This trace describes the effect of the action on the environment and whether it was executed successfully, and it is provided as additional context to the high-level policy at the next timestep.

For both mobile-use and computer-use domains, we use GPT-4o (OpenAI et al., 2024) as both the planner and reflection model. We use UI-TARS-1.5 7B (Qin et al., 2025) as the grounding model for mobile and GTA1-7B (Yang et al., 2025) as the grounding model for computer-use. In the computer-use setting, we additionally supply the high-level policy with one heuristic action, to enable interaction with complex UI elements (like spreadsheets, detailed in Appendix F), which improves data collection success rates. These heuristic actions are only used by the expert and are not available to the trained AUTOPLAY policy.

**Task Trajectory Verifier**. To determine whether AUTOPLAY-generated tasks are executed successfully, we employ a task verifier that evaluates trajectories. The verifier is an MLLM that takes as input the task instruction and the executed trajectory (represented as interleaved images and actions). It operates in three stages: (i) summarizing what the agent is doing in the trajectory, (ii) producing a Chain-of-Thought (Wei et al., 2023) reasoning for whether the task has been completed, and (iii) issuing a final judgment of "success" or "failure". We use GPT-4o as the task verifier for trajectories collected using the task executor agent. Full details of the verifier and prompts are in Appendix C.4.

**SFT Data Generation**. To generate SFT data for mobile-use domain, we use 20 mobile apps from Rawles et al. (2024). Similarly, for computer-use domain, we use 13 desktop apps from Xie et al. (2024) on Ubuntu devices. We run the environment exploration for 3 turns to build the environment context and we define 4 task guideline prompts. Using these, we generate 50 tasks for each app, task guideline prompt, exploration context combination. We specify the precise prompts used for the task guidelines in Appendix C.1 for each platform. We are able to generate $\sim 20k$ tasks for Android platform and $\sim 10k$ for Ubuntu platform. After execution of these tasks and subsequent verification with the verifier we obtain $\sim 8k$ and $\sim 3.5k$ successful trajectories respectively to use for SFT.

**Base Model**. For our experiments, we use Qwen2.5-VL Instruct (Bai et al., 2025) 3B-72B MLLMs as the base model and finetune them using SFT and RL on our dataset to build a end-to-end UI agents. At each timestep, the policy takes as input: current image observation, history of past actions, the task instruction and outputs low-level actions. Appendix F describes the action space per-domain.

### 3.4 REINFORCEMENT LEARNING ON SYNTHETIC TASKS

We also use the tasks generated by AUTOPLAY in conjunction with the task verifier to scale reinforcement learning (RL) training of mobile-use agents. For each RL training environment worker, we sample a random task from the AUTOPLAY task dataset $(g, s_0) \sim \mathcal{D}$. We then initialize the simulator state to $s_0$ and task our agent to successfully complete the instruction. At the end of the rollout, we score the trajectory with the task verifier as either a success or failure, assigning a reward of 1 or 0 to the trajectory respectively. We use the Qwen2.5-VL-32B-Instruct Bai et al. (2025) MLLM to instantiate the task verifier using the same verifier setup as in Sec. 3.3. By leveraging the AUTOPLAY task generator and an MLLM task verifier we unlock RL training with verifier feedback without requiring any human annotation. RL allows training on all AUTOPLAY tasks, even those the executor is not able to solve. However, for better training stability, we restrict RL training to the $8k$ tasks the executor can successfully solve at least once. We use GRPO to train the model with RL with group size 8 across 32 H100 GPUs. See Appendix B.2 for full details.

## 4 EXPERIMENTS

In this section, we demonstrate that AUTOPLAY enables scaling generation of synthetic tasks for UI agents, that are feasible, diverse and grounded in environment state and functionality all without human annotations. The resulting high-quality tasks enable scaling of supervised finetuning (SFT) and reinforcement learning (RL) for post-training MLLMs as capable UI agents. We also show

**Table 1(a) Android World**

| Method
Pass@5 | Model Size | Pass@1 |
|---|---|---|
| GPT4o + UI-TARS | 43.1 | 55.3 |
| Seed-VL 1.5 | 62.1 | – |
| UI-TARS 2 230B | 73.3 | – |
| UI-TARS 1.5 7B | 26.4 | 36.2 |
| UI-TARS 72B | 37.7 | 53.0 |
| Qwen-VL 2.5 7B | 19.5 | 27.7 |
| Qwen-VL 2.5 72B | 35.0 | 43.5 |
| AUTOPLAY-7B | 40.1 (+20.6 △) | 58.4 (+30.7 △) |
| AUTOPLAY-72B | 47.9 (+12.9 △) | 68.2 (+24.7 △) |
| AUTOPLAY-3B | 34.2 | 52.2 |
| AUTOPLAY-3B RL | 39.9 (+5.7 △) | 53.4 (+1.2 △) |

**(a) Android World**

**Table 1(b) OSWorld**

| Method | Pass@1 | Pass@5 |
|---|---|---|
| o3 + GTA1 | 34.0 | – |
| AguVis 72B | 10.3 | – |
| UI-TARS-1.5 7B | 24.5 | – |
| UI-TARS-1.5 72B | 42.5 | – |
| Qwen-VL 2.5 7B | 3.7 | 4.1 |
| Qwen-VL 2.5 72B | 4.4 | 5.4 |
| AUTOPLAY-7B | 11.4 (+7.7 △) | 12.1 (+8.0 △) |
| AUTOPLAY-72B | 14.5 (+10.1 △) | 16.0 (+10.6 △) |

**(b) OSWorld**

**Table 1.** Evaluation results on UI agent benchmarks. Pass@1 and Pass@5 values for AUTOPLAY models include in parentheses the change relative to the corresponding base model of the same parameter size.

the tasks synthesized using AUTOPLAY achieve a higher task execution success rate and enable training more capable downstream agents compared to methods that do not explicitly explore the environment (Zhou et al., 2025) and ones that use iterative exploration (Xie et al., 2025; Trabucco et al., 2025). Furthermore, we also evaluate importance of *environment exploration* and *task guidelines* for task generation and find that both components are important to build a effective task proposer.

## 4.1 AUTOPLAY DATA FOR UI AGENTS

To assess how effectively AUTOPLAY generates training tasks for UI agents, we measure the performance of agents trained with these tasks on established downstream benchmarks: Android-World (Rawles et al., 2024) for mobile agents and OSWorld (Xie et al., 2025) for desktop agents. We use the AUTOPLAY data described in Section 3.3 to train a separate agent per benchmark. In both benchmarks, the agent is evaluated using ground truth success verifier that has access to privileged environment information. We report the average success rate, referred to as Pass@1, over 5 independent random trials since most benchmark tasks have stochastic starting states with different goal details and application content. We also report the Pass@5 metric, which is the percentage of tasks where *any* of the 5 independent trials succeeds.

**AUTOPLAY improves agentic capabilities of base models**: We use the synthetic data to finetune a base Qwen-2.5-VL-XB into an agentic model AUTOPLAY-XB for a range of sizes $X \in \{3, 7, 72\}$. The resulting models consistently outperform their base as shown in Table 1. For example, AU-TOPLAY-7B outperforms its base, Qwen2.5-VL-7B, by 20.6% on AndroidWorld and by 7.7% on OSWorld. These gains demonstrate that the tasks generated by AUTOPLAY are highly relevant to the broad spectrum of UI agent tasks evaluated in both benchmarks, despite requiring *no human data* collection. Similarly, the largest model, AUTOPLAY-72B, surpasses the strong Qwen2.5-VL-72B baseline by 12.9% points on AndroidWorld and 10.1% points on OSWorld, illustrating that even highly capable UI agents benefit substantially from AUTOPLAY 's synthetic tasks. Remarkably, AUTOPLAY-3B achieves a 34.2% success rate on AndroidWorld, nearly matching the performance of the much larger Qwen2.5-VL-72B model, which attains 35.0%.

**AUTOPLAY achieves competitive results with Proprietary Baselines trained on human data**: Table 1 further shows that AUTOPLAY outperforms strong UI agent models such as UI-TARS-1.5 (Qin et al., 2025), which was trained on large-scale human-annotated GUI data, by 10.2% at the 7B scale and 13.7% at the 72B scale on AndroidWorld. On OSWorld, AUTOPLAY-72B surpasses the two-stage training pipeline used in AguVis-72B (Xu et al., 2024), which leverages both grounding data and human-annotated GUI navigation data, by 5.0% in success rate. Although UI-TARS-2 230B and Seed-VL-1.5 (ByteDanceSeedTeam, 2025) achieve higher performance on AndroidWorld, they rely on substantially larger mixtures of expert models. Similarly, UI-TARS outperforms AUTOPLAY on OSWorld, likely due to its use of curated, human-labeled UI data—whereas AUTOPLAY autonomously explores, proposes tasks, and collects data without human supervision.

**AUTOPLAY eventually outperforms the Executor**: We also find in Table 1 that AUTOPLAY-72B outperforms the GPT-4o + UI-TARS (OpenAI, 2024; Qin et al., 2025) policy used to collect the data from the generated tasks by 4.8% on the average success rate. This demonstrates the strength of the task verifier for automatically filtering successful trajectories, without ground truth environment

| Task Generator | TASK EXECUTION | ANDROID WORLD | |
| --- | --- | --- | --- |
| | **Pass@1** ($\uparrow$) | **Pass@1** ($\uparrow$) | **Pass@5** ($\uparrow$) |
| No Exploration | 21.3 | 28.8 $_{\pm 1.5}$ | 49.2 |
| Iterative Exploration | 56.4 | 21.6 $_{\pm 1.7}$ | 33.6 |
| AUTOPLAY w/o task guidelines | 43.5 | 26.7 $_{\pm 2.7}$ | 38.9 |
| AUTOPLAY | 46.0 | 38.2 $_{\pm 3.1}$ | 58.5 |

**Table 2. Ablations**: We compare AUTOPLAY-7B with two baselines that perform lesser exploration, *No Exploration* and *Iterative Exploration*, as well as AUTOPLAY-7B without using *task guidelines*.

information. Furthermore, AUTOPLAY-72B achieves 12.9% higher Pass@5 than the data collection policy, indicating it's ability to solve new tasks, not just more robustly solve the same tasks.

**AUTOPLAY is effective for RL training**: In Table 1 we also highlight that it is feasible to perform RL training on the AUTOPLAY generated tasks, according to the process described in Section 3.4. We see a 5.7% gain in AndroidWorld. With RL training, the AUTOPLAY-3B model performs similarly to the AUTOPLAY-7B model trained with just SFT (39.9% versus 40.1% success rate).

## 4.2 ABLATION ANALYSIS

Next, we look at the importance of individual design decisions in AUTOPLAY. We use AndroidWorld and generate 5,000 Android tasks for all models. We then apply the same executor and verifier described in Section 3.3 to obtain successful trajectories, which are used to fine-tune the Qwen2.5-VL-7B model with supervised learning for AUTOPLAY and variations. Full details are in Appendix E.

**Exploration Ablations:** To quantify the importance of the environment exploration we compare with two alternative task proposal baselines. The first baseline, *No Exploration*, is based on Zhou et al. (2025); Xie et al. (2025) and, unlike AUTOPLAY, performs no exploration of the domain. Instead, it generates tasks solely from static environment context, such as textual descriptions and application starting screenshots. To implement *No Exploration*, we manually write detailed descriptions of each AndroidWorld app's features and task guidelines in the same style as those used for AUTOPLAY. The second baseline, *Iterative Exploration*, follows Pahuja et al. (2025); Xie et al. (2025) and incorporates limited exploration. However, it only attempts to sequentially execute a series of short-horizon subgoals and summarize them as tasks, without performing broad exploration of the domain. As a result, it is more constrained than AUTOPLAY. Our implementation begins at the home page of the target application. An MLLM proposes a short-horizon subgoal based on the initial screenshot, and the same data collection policy used in AUTOPLAY executes it. The MLLM then proposes the next subgoal, and this process repeats up to 7 times. Finally, a summarizer MLLM takes as input the textual descriptions of the executed subgoals, along with their success or failure, and produces a long-horizon goal for the full trajectory—akin to hindsight relabeling.

**AUTOPLAY generates more feasible tasks:** Out of the 5$k$ tasks generated by each method, Table 2 shows the same executor succeeds in 46.0% of the AUTOPLAY tasks compared to only 21.3% for *No Exploration* and 56.4% for *Iterative Exploration*. This demonstrates that more tasks generated by AUTOPLAY are feasible. A common failure mode of *No Exploration* baseline was hallucination in the proposed task since it only relies on app functionality description. AUTOPLAY, on the other hand, uses exploration context to ground task details in the actual states of the environment.

**AUTOPLAY tasks train better agents:** Table 2 also shows that agents trained with AUTOPLAY tasks outperform agents trained with tasks from *No Exploration* and *Iterative Exploration*. Agents trained with AUTOPLAY tasks outperform those trained with *No Exploration* tasks by 9.4% on Pass@1 and Pass@5 success rates on AndroidWorld. AUTOPLAY tends to cover a broader range of functionalities in the environment compared to *No Exploration*. While *Iterative Exploration* does interact with the environment to generate tasks, AUTOPLAY tasks train agents that are 16.6% more successful, as shown by Task Execution Pass@1. This is because *Iterative Exploration* synthesizes long horizon trajectories by stitching easier short horizon subgoals to guide exploration. This consequently leads to less diverse and easier tasks. In contrast, through rounds of long-horizon exploration, AUTOPLAY generates diverse tasks that provide broad coverage over app functionalities.

**Exploration generates more diverse tasks**: We compare the task distributions generated by AUTOPLAY and *No Exploration* in Figure 3. The distribution is computed over manually defined task categories that cover a broad range of possible tasks. For example, tasks in the *Composition* category

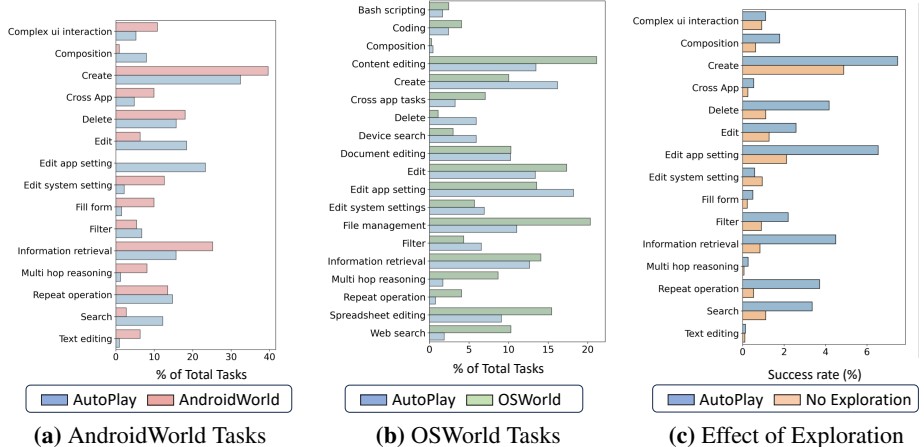

**Figure 3. Task Coverage**. **Left and Middle:** For a set of predefined categories, we compare the task distribution across AndroidWorld/OSWorld test tasks and AUTOPLAY generated tasks in light color. **Right:** For a set of predefined categories, we compare the task distribution across AUTOPLAY generated tasks and *No Exploration* generated tasks in light color. Additionally, we show number of tasks that AUTOPLAY-7B executor solves for both task sets in darker color bars.

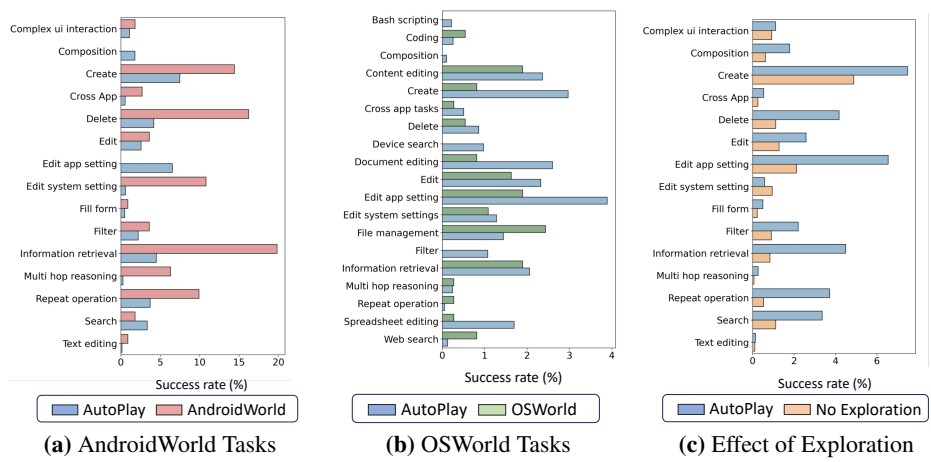

**Figure 4. Task Success Rates**. **Left and Middle:** For a set of predefined categories, we compare number of tasks that AUTOPLAY-7B executor solves per task category across AndroidWorld/OSWorld test tasks and AUTOPLAY generated tasks. **Right:** For a set of predefined categories, we compare number of tasks that AUTOPLAY-7B executor solves per task category across AUTOPLAY generated tasks and *No Exploration* generated tasks.

require combining multiple skills or subtasks to achieve the overall goal (e.g., "Find when John is free and schedule a meeting with him for this week."). Additional details about the task categories are provided in Appendix D. Although the two distributions exhibit similar trends—categories that are more prevalent under one method tend to be prevalent under the other—the results consistently show lower execution success rates for *No Exploration* compared to AUTOPLAY, particularly in categories such as deleting, editing, or retrieving in-app data. This highlights the importance of grounding task generation in exploration.

**Task guideline ablation:** The task guidelines prompts steer the AUTOPLAY task generator towards diverse categories with good domain coverage. Table 2 ablates the impact of the task guideline prompts and illustrates they are important for generating tasks that train performant agents. We see a minor boost in the ability of the executor to solve tasks, showing that task guidelines improve task feasibility. More importantly, guidelines provide a substantial boost in the downstream performance.

| App Name | TRAIN ON HOLDOUT APP | | HOLDOUT APP | |
|---|---|---|---|---|
| | Pass@1 (↑) | Pass@5 (↑) | Pass@1 (↑) | Pass@5 (↑) |
| Markor | 26.1 | 38.4 | 11.5 | 38.5 |
| Simple Calendar Pro | 33.6 | 70.5 | 26.8 | 47.1 |
| Broccoli-Recipe Expense | 20.0 | 38.6 | 35.4 | 46.1 |
| Pro Expense | 47.7 | 66.6 | 51.1 | 66.6 |
| Settings | 74.4 | 80.0 | 72.0 | 80.0 |
| Average Performance | 40.4 | 58.9 | 39.4 | 55.7 |

**Table 3. Generalization Results**: Evaluation results of AUTOPLAY-7B to holdout apps *vs.* training on test app.

**AUTOPLAY generated tasks cover platform functionality**: In addition, we compare of how well AUTOPLAY's task distribution mimicks a natural task distribution. For this, we use AndroidWorld and OSWorld benchmarks as representative task distributions. As shown in Figure 3, AUTOPLAY produces task covering the majority of the categories. Further, our method seems to follow the distributions of these benchmarks. As shown in Figure 4, for AndroidWorld, the task executor performs quite well on majority of the categories except tasks where skills like fine-grained text editing, complex UI interactions (*e.g.* date-time picker wheels, etc), and cross app navigation is required which leads to the downstream policy perform poorly on these task categories. This clearly suggests, improving the task executor while keeping the task generator the same would lead to better quality synthetic dataset to improve downstream agent performance. For OSWorld, AUTOPLAY covers common task categories like document editing, however, it struggles to generate tasks with cross-app interaction, bash scripting and web search. We attribute this gap in task coverage to lack of sufficiently diverse task guidelines prompts for computer-use domain.

**Generalization to unseen environments**. In this section, we answer the question: How well do agents trained using AUTOPLAY data generalize to unseen environments compared to training on test environments? To study this question we holdout 5 mobile applications (Settings, Markor, Simple Calendar Pro, Pro Expense, Broccoli-Recipe App) from our training split and evaluate generalization to these apps zero-shot at test time. Due to limited number of environments we train 5 separate models while holding out data from a single environment. In addition, we also train a model only on data collected in the held out environment. We compare the results of zero-shot generalization to held out environment with agents trained only on data from held out environment in Tab. 3. We find that agents trained on larger datasets generalize comparably well to holdout environments compared to training agents on data collected in test environments ($39.4\%$ *vs.* $40.4$ on Pass@1 success rate).

**Verifier analysis**. In Tab. 4, we present a human evaluation study to validate effectiveness and human agreement of our verifier. We find the verifiers we used for SFT and RL show ∼60% accuracy compared to ground truth labels and have high recall. Specifically, we sampled 200 unique tasks with ∼50% execution success rate and asked 3 human annotators to label them. We label a demonstration as successful if at least 2 annotators agree on it. Next, using human labels as ground truth we compare accuracy, precision and recall for our GPT-4o and Qwen-2.5 VL Instruct model based verifiers in table below. We find both verifiers achieve ∼60% accuracy and have high recall but low precision. This suggests these models are more optimistic than pessimistic in labelling a task as successful. We find the key failure mode of these models is to mark a long horizon task as successful when an agent completes partial subgoals but doesn't necessarily complete the full task.

## 5 CONCLUSION

We introduce AUTOPLAY, a scalable pipeline for synthesizing diverse, feasible, and verifiable tasks for post-training MLLM-based interactive agents. AUTOPLAY achieves this by explicitly exploring interactive environments combined with a carefully curated task guideline prompts to ground task generation in discovered states and accessible functionalities. We demonstrate the effectiveness of AUTOPLAY in training UI agents across both mobile and computer domains. AUTOPLAY generates 20k tasks across 20 Android applications and 10k tasks across 13 applications Ubuntu applications to train mobile-use and computer-use agents. This dataset enables training MLLM-based UI agents that improve in success rates up to $20.0\%$ on mobile-use and $10.9\%$ on computer-use domains. Furthermore, AUTOPLAY generated tasks combined with MLLM reward models enable scaling reinforcement learning training of UI agents, leading to an additional $5.7\%$ gain. Overall, these results establish AUTOPLAY as a scalable approach for post-training MLLM agents, reducing reliance on human annotation while significantly enhancing agent performance across domains.

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

| Verifier Model | Accuracy (↑) | Precision (↑) | Recall (↑) |
|---|---|---|---|
| GPT-4o | 59.3 | 56.9 | 82.8 |
| QwenVL-2.5 Instruct | 61.8 | 57.7 | 95.9 |

**Table 4. Verifier analysis**: Human evaluation of verifier accuracy on AndroidWorld data.

| Verifier | Pass@1 (↑) | Pass@5 (↑) | Dataset Size |
|---|---|---|---|
| 1) GPT-4o | 40.1 | 58.4 | $8k$ |
| 2) Qwen2.5-VL 72B | 38.6 | 54.5 | $12k$ |

**Table 5. Verifier Ablations**. Evaluation results of training Qwen-VL 2.5 7B using SFT with different verifiers on AndroidWorld.

## A    ADDITIONAL EXPERIMENTS

**Impact on distillation performance with different verifiers**. In this section, we ablate the choice of verifier model used for data filtering for SFT and its impact on downstream performance. Specifically, we compare use of proprietary MLLMs like GPT-4o *vs.* open-source MLLM Qwen2.5-VL Instruct. This experiment is especially important to evaluate what capability level is required in a MLLM to be used as a effective verifier and whether smaller open-source MLLMs can be used as effective verifiers to enable RL scaling as proprietary models cannot be used due to cost implications. To do so, we use all $\sim 20k$ tasks and demonstrations generated by our synthetic task proposer and executor and verify these using both GPT-4o and Qwen2.5-VL Instruct 72B to create two SFT datasets. In Tab. 5, we present results of finetuning Qwen2.5-VL Instruct 7B model on these trajectory datasets and present evaluation results on AndroidWorld benchmark. We find that model trained on GPT-4o verified trajectories outperforms Qwen2.5-VL Instruct 72B verified trajectories,

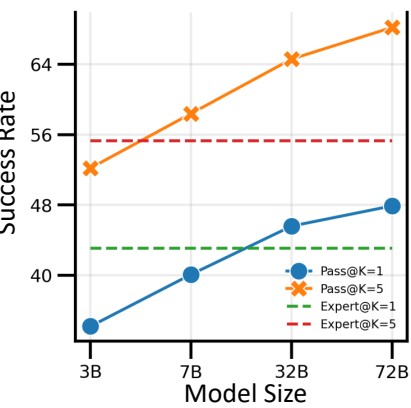

**Figure 5.** Effect of increasing model size in AUTOPLAY.

however, the difference in performance is quite small $1.5\%$ suggesting smaller open-source models are capable verifiers that can enable RL training in mobile-use domains.

### A.1    GENERALIZATION RESULTS

So far for all experiments we have used same set of training and test environments. Even though these experiments demonstrate effectiveness of our pipeline in providing a broad coverage over training task distribution and its impact on downstream performance, another important question is: How well do agents trained using synthetic data generalize to unseen environments compared to training on test environments? To study this question we holdout 5 mobile applications (Settings, Markor, Simple Calendar Pro, Pro Expense, Broccoli-Recipe App) from our training split and evaluate generalization to these apps zero-shot at test time. Due to limited number of environments we train 5 separate models while holding out data from a single environment. In addition, we also train a model only on data collected in the held out environment. We compare the results of zero-shot generalization to held out environment with agents trained only on data from held out environment in **??**. We find that agents trained on larger datasets generalize comparably well to unseen (or test) environments compared to training agents on data collected in test environments.

# B  TRAINING DETAILS

## B.1  ALGORITHM PARAMETERS

We provide parameter values for AUTOPLAY, as listed in Algorithm 1, for both AndroidWorld and OSWorld experiments:

| Parameter Name | AndroidWorld | OSWorld |
|---|---|---|
| $S$ # of apps | 20 | 13 |
| $K$ # of exploration turns | 3 | 5 |
| $P$ # of task guidelines | 4 | 4 |
| $K$ # of task generations per guideline and context | 50 | 50 |

## B.2  RL TRAINING DETAILS

We use GRPO (Shao et al., 2024) for our RL finetuning experiments using 32 H100 GPUs. As we require a MLLM verifier for evaluating task success and reward generation we use 2 GPUs on everywhere GPU node to host the Qwen2.5-VL Instruct 32B (Bai et al., 2025) MLLM using vLLM (Kwon et al., 2023) inference engine. The verifier only takes the final 8 frames of task execution as input and predicts a binary evaluation of the task execution in addition to reasoning and task summary as described in Sec. 3.3.

Each GPU node runs environment workers and RL trainer on 6 of the GPUs with 8 environments per GPU. We evaluate the policy after 120 GRPO updates. This is equivalent to $8 \times 24 \times 16 \times 120 = 368,640$ environment samples. Additional hyperparameters are detailed in Tab. 6.

# C  ADDITIONAL AUTOPLAY DETAILS

## C.1  TASK PROPOSER AGENT

AUTOPLAY task proposer agent operates in two stages where it first explores the environment in a goal-agnostic manner to collect experience from the environment that can aid in synthesis of high-quality task that are grounded in environment state and feasible to execute. The task proposer takes as input the task proposer prompt, goal-agnostic exploration observations and task guideline prompt to generate a sequence of tasks. We present the task proposer prompt prefix in Appendix C.1.

Next, for mobile-use domain we define 4 separate task guideline prompts that are specific to mobile-use domain which the task proposer uses in addition to exploration experience to generate tasks. These prompts cover tasks of following types: (1.) Feature-Use Appendix C.1: Tasks that use basic features of the app and provide broad coverage over all features shown, (2.) Feature-Composition Appendix C.1: Tasks that composes multiple feature-use tasks to create complex tasks with multiple subtasks, (3.) Information Retrieval Appendix C.1: Tasks that require searching for specific information requiring or answering queries asked by users about state of the environment, (4.) Feature-Repetition Appendix C.1: Tasks that require executing feature-use tasks over multiple entities stored in an application (*e.g.* deleting multiple calendar events).

For computer-use domain we define 2 separate task guideline prompts mentioned in Appendix C.1 and Appendix C.1.

---

**Task Generator MLLM Prompt (Mobile-Use: Android and Computer-Use: Ubunt)**

```
You are a capable UI understanding agent. You will be provided a set of images from the {PLATFORM}
app that shows different features and the current state of the app. Your task is to convert the
described functionalities into a list of tasks that a useful UI understanding agent should be able
to complete. Use the images from the app to ground these tasks to the ones that are feasible.
Propose as many diverse tasks as possible to cover broad range of features.

For all described and demonstrated functionalities output a list of up to {NUM_TASKS} unique tasks
that can be executed in the app as a JSON which should be in the following format:
```

| Parameter | Value |
|---|---|
| Number of GPUs | 24 |
| Number of environments per GPU | 8 |
| Rollout length | 16 |
| GRPO Group Size | 8 |
| Number of mini-batches per epoch | 4 |
| LR | $1.e^{-6}$ |
| Entropy coefficient | 0.0 |
| KL divergence coefficient | 0.0 |
| LR scheduled | 10 linear warmup steps from 0 LR to 1e-6 LR |
| Max gradient norm: 1.0 | |
| Optimizer | Adam |
|    Weight decay | 0.0 |

**Table 6.** Hyperparameters used for RL finetuning using GRPO.

```
tasks = [
    {
        "thought": "<Detailed thought and reasoning for the proposed task, why is the task simple
        enough to execute and whether it satisfies the task proposal guidelines>",
        "instruction": <natural language instruction describing a task with/without template
        params with name of the app>,
        "tag": <few words describing the type of task>,
        "app_name": <name of the app>,
        "template params": {
            "param_name": {
                "description": <param description>,
                "possible_values": [<list of 5 random values>]
            }
        }
    }
]

{TASK_GUIDELINE_PROMPT}

{ENVIRONMENT_CONTEXT}
```

**Feature-Use Task Guidelines Prompt (Mobile-Use: Android)**

```
For each functionality/feature make sure to follow these guidelines while generating tasks:
1. Create all possible tasks. For example, if there is a clock app on the phone which has
worldwide clock feature then you can create tasks like: "Add worldwide clock for <city>", "Remove
worldwide clock for <city>", "Add worldwide clock for <city_1, city_2> and reorder to put <city_2>
before <city_1>". Make sure to add the <param> details in "template params" in JSON to allow
creation of diverse tasks.
2. In any instruction if there are going to be templated parameters then add it in following
format: {param_name}
3. If a task requires creating/adding/editing/deleting any information/event/entry make sure all
required entities for completing the task are parameterized/templated. For example:
    a.) For a task that requires creating notes both the name of the new note and content of the
    note should be specified as parameters.
    b.) Similarly, if a task requires editing some details describe the exact edit you'd like to
    make, what entity needs to be edited. When proposing such tasks, only ask edits to existing
    entities shown in the images from the app for such tasks. If no such entities exist ask the
    agent to first create one and then edit it.
    c.) If a task requires deleting an entity, describe the entity that needs to be deleted. When
    proposing such a task, only ask to delete an entity that already exists as shown in the images
    from the app for such tasks. If no such entities exist ask the agent to first create one and
    then delete it.
    d.) If a task requires copying an entity, describe the entity that needs to be copied and
    where it should be copied to (ex: which folder or date). When proposing such a task, only ask
    to copy an entity that already exists as shown in the images from the app for such tasks. If
    no such entities exist ask the agent to first create one and then copy it.
    e.) For all parameters that are templated if any parameter refers to an entity for edit,
    delete or copy task make sure that the list of possible values only contain entities that
    exists or will be created.
4. Ensure that instructions are unambiguous and clearly describes a actual task that can be
performed on the app.
```

```
5. Do not include tasks that require accessing/uploading/capturing content from real world. For
example, scanning a document, recording a new video, taking a picture using camera.
6. Task instructions should be natural user requests someone might actually ask a capable UI agent
to complete on the app.
7. If any supports browsing information from the web or library make sure to include that in the
task instruction. For example, if a task requires searching for a book on the web, make sure to
include that in the task instruction.
```

### Information Retrieval Task Guidelines Prompt (Mobile-Use: Android)

```
For each proposed task make sure to follow these guidelines:
1. Create tasks that require retrieving different types of information that are useful for a user
to make decisions. These tasks should be natural user requests someone might ask in real world.
For example, if there is a calendar app on the phone you can propose tasks like: "How many
meetings do I have scheduled for {date}?", "What meetings do I have to attend between {start_time}
and {end_time} on date {date}? List each event separated by comma". Make sure to add each
templated param {param} in "template params" in JSON to allow creation of diverse tasks.
2. Task instructions should be natural user requests someone might actually ask a capable UI agent
to complete on the app. For example, "Check the current sample rate set in the Audio Recorder
app." is a bad task because of how it is specified. Instead, propose tasks like "What is the
current sample rate set in the Audio Recorder app?" or "Can you tell me the current sample rate
set in the Audio Recorder app?". The latter two tasks are better because they explicitly ask for
the information to be retrieved and are more natural user requests.
3. For all such tasks, also generate a "answer" field in the task JSON which contains a natural
language answer to the task. This answer should be a valid answer that can be retrieved from the
app. For example, if the task is "What is the weather forecast for {city} on {date}?", the answer
should be a valid weather forecast for the specified city and date.
4. For any task instruciton if there are going to be templated parameters then add it in following
format: {param_name}
5. For all such task propose different varieties of information retrieval tasks that require
searching for information in the app and covers different features. Here are some examples for
different apps:
    a.) For a notes app, you can ask "How many todos I have listed in the notes app?", "What todos
    are pending for today in the list from note app?",etc.
    b.) For a fitness app, "What workouts do I have planned for this week?", "How long did I
    workout in last week?", etc.
6. Only proposes tasks where the last action the agent should require to solve the task is to
return a natural language response with the information requested to the user. For example, "What
is the weather forecast for {city} on {date}?" is a good task because it requires the agent to
return a natural language response with the weather forecast for the specified city and date.
Whereas, "How do I view the weather forecast?" or "How can I set the city to {city} in weather
app?" is not a good task because it does not specify the information to be retrieved and requires
the agent to show a demonstration in the app to view the weather forecast.
7. Do not propose ambiguous tasks that do not specify the information to be retrieved. For
example, "What is the weather like?" is too generic and should be avoided. Instead, propose tasks
like "What is the weather forecast for {city} on {date}?".
8. If any app supports browsing information from the web or library make sure to include tasks
that require searching information from web. For example, "How much is the price of a book titled
{book}?", etc.
```

### Feature Composition Task Guidelines Prompt (Mobile-Use: Android)

```
For each task proposed make sure to follow these guidelines:
1. Each task should be a composition of multiple subtasks that require a UI agent to execute
sequence of subtasks across multiple functionalities/subtasks. For example, if there is a clock
app on the phone which has worldwide clock, alarms, and timer feature then you can create tasks
like: "Add worldwide clock for {city} then set an alarm for {time} on days {}", "Remove worldwide
clock for {city} and set a timer for {hour} hours, {minute} minutes", "Delete the alarm for {time}
and delete all the world clocks". Make sure to add each templated {param} details in "template
params" in JSON to allow creation of diverse variants of each tasks.
2. Task instructions should be natural user requests, unambiguous and clearly describes a actual
task that can be performed on the app. For example, "Create a calendar event for meeeting titled
{title} at {time} on {date} for duration {duration} for meeting with Bob then delete the first
event on {date2}" is a good task as it specifies all required details. In contrast, a task like
"Create a calendar event for titled {title} on {date} then delete the first event on {date2}" is a
bad task as it does not specify the start time/duration of the event.
2. For any instruction if there are going to be templated parameters then specify it in following
format in the instruction: {param_name}
3. For all such tasks that require creating/adding/editing/deleting any information/event/entry
make sure all required entities for completing the task are parameterized/templated. For example:
    a.) For notes app, "Create a note titled {note_title} with content {note_content} in {folder}
    folder and then delete note titled {note_title_2}" is a good task as it templates the title,
    content, and location of notes.
```

```
    b.) For expense app, "Create expenses for {expense_name_1}, amount {amount_1}, category
    {category_1}, note {note_1} followed by expense {expense_name_2}, amount {amount_2}, category
    {category_2}, note {note_2} and then delete all duplicate expenses".
    c.) For files app, you can ask "Search for file named {name_1} and then go delete the files
    named {name_2}, {name_3}".
4. Do not include tasks that require accessing/uploading/capturing content from real world. For
example, scanning a document, recording a new video, taking a picture using camera.
5. If any app supports browsing information from the web or library make sure to include that in
the task instruction. For example, if a task requires searching for a book on the web, make sure
to include that in the task instruction.
```

### Subtask Repetition Task Guidelines Prompt (Mobile-Use: Android)

```
For each task proposed make sure to follow these guidelines:
1. Each task instruction you propose should repeatedly ask to execute the same feature or subtask
for a single functionality. For example, if there is a calendar app on the phone which has
multiple calendar events then you should propose tasks like: "Delete events {event_1}, {event_2},
{event_3}", "Delete all events on date {date_1} {date_2}", "Delete events {event_1} on {date_1},
{event_2} on {date_2}, {event_3} on {date_3}" and so on. Make sure to add each templated {param}
details in "template params" in JSON to allow creation of diverse variants of each tasks. Make
sure the for each instance of event in template params unique values from screenshots shown are
used.
2. Task instructions should be natural user requests, unambiguous and clearly describes a actual
task that can be performed on the app. The instructions can specify same task in different ways.
For example, "Delete all events on {date_1}" can also be specified as "Delete all events on {day}
of the {week}", and a task like "Delete all events on this weekend" can be specified as "Delete
all events on Saturay and Sunday of current week".
3. Also specify tasks in various ways that require inferring details of the entity being referred
by looking at details from the screenshot. For example, if the app is a expense app you can
specify tasks like "Delete expenses that have expense amount greater than {amount}" or "Delete all
expenses that are in {category} category" or "Delete all expenses that have a note containing
{note_content}".
4. Propose tasks that require repeatedly executing same feature for all types of tasks like
create/delete/edit. For example, if the app is a notes app you can propose tasks like "Create
notes titled {note_1}, {note_2}, {note_3} with content {content_1}, {content_2}, {content_3}",
"Delete notes titled {note_1}, {note_2}, {note_3}", "Edit notes titled {note_1} to change content
to {new_content_1}, edit note titled {note_2} to change content to {new_content_2}" and so on.
5. For any instruction if there are going to be templated parameters then specify it in following
format in the instruction: {param_name}
6. Do not include tasks that require accessing/uploading/capturing content from real world. For
example, scanning a document, recording a new video, taking a picture using camera.
7. If any app supports browsing information from the web or library make sure to include that in
the task instruction. For example, if a task requires searching for a book on the web, make sure
to include that in the task instruction.
```

### Feature-Use Task Guidelines Prompt (Computer-Use: Ubuntu)

```
For each functionality/feature make sure to follow these guidelines while generating tasks:
1. Create all possible tasks that use features shown by the demonstrations. For example, if there
is a code IDE app on the desktop then you can propose tasks like: "Create a new project at path
<path> from UI or terminal", "Install an extension <extension_name> for auto formatting text",
"Change the settings of my editor to set line-length to <value>", etc. Make sure to add the
<param> details in "template params" in JSON to allow creation of diverse tasks.
2. In any instruction if there are going to be templated parameters then add it in following
format: {param_name}
3. If a task requires creating/adding/editing/deleting any information/event/entry make sure all
required entities for completing the task are parameterized/templated. For example:
    a.) For a task that requires creating a new file in a coding IDE specify the name of the file,
    location, and content of the file as parameters. Similarly, if a task requires creating a new
    project specify the name of the project, location, and any other required details as
    parameters.
    b.) If a task requires editing some details describe the exact edit you'd like to make, what
    entity needs to be edited. When proposing such tasks, only ask edits to existing entities
    shown in the images from the app for such tasks. If no such entities exist ask the agent to
    first create one and then edit it. For example, if a task requires editing a image in image
    editing app, specify the image to be edited, the edit to be made, and any other required
    details as parameters.
    c.) If a task requires deleting an entity, describe the entity that needs to be deleted. When
    proposing such a task, only ask to delete an entity that already exists as shown in the images
    from the app for such tasks. If no such entities exist ask the agent to first create one and
    then delete it.
    d.) If a task requires copying an entity, describe the entity that needs to be copied and
    where it should be copied to (ex: which folder or date). When proposing such a task, only ask
    to copy an entity that already exists as shown in the images from the app for such tasks. If
    no such entities exist ask the agent to first create one and then copy it.
```

```
    e.) For all parameters that are templated if any parameter refers to an entity for edit,
    delete or copy task make sure that the list of possible values only contain entities that
    exists or will be created.
4. Ensure that instructions are unambiguous and clearly describes a actual task that can be
performed on the app.
5. Task instructions should be natural user requests someone might actually ask a capable UI agent
to complete on the app.
6. If any supports browsing information from the web or library make sure to include that in the
task instruction. For example, if a task requires searching for a book on the web, make sure to
include that in the task instruction.
```

---

**Feature Composition Task Guidelines Prompt (Computer-Use: Ubuntu)**

```
Available Primitives:
- search: searches for something using the search bar. You can optionally filter the results based
on the given criteria.
- filter: filters the results based on the given criteria. You can compose multiple filters to
form a single filter. A filter can be direct, or it needs to be inferred using multi_hop_reasoning.
- edit: edit/modifies a property (changing to celsius, sorting results, comparing, modifying the
view by clicking on a button, etc.)
- delete: deletes something from the page. can also be used to delete something from the cart.
- add: add something to the cart. You can combine multiple add primitives, you can also add
multiple quantities of the same item.
- multi_hop_reasoning: criteria for selection/filtering is not mentioned directly, but requires
reasoning to be performed. Examples: "one month from now", "30\% of an income of \$8000".
- repeat: repeats the results based on the given criteria. Example: add red wine, white wine. Add
2 bottles.
- navigate: navigates to the given URL, click on a link to navigate to its page. select one of the
entries from a list.
- form: fills out a form (contact, login, enter numbers to perform a calculation, filling in zip
code, etc.). We can stack multiple forms to create a task.
- and: logical operation

For each functionality/feature make sure to follow these guidelines while generating tasks:
1. Generate 10 unique tasks each with 2, 4, 6 primitives in your task composition (minimum 3,
maximum 6). Make sure the tasks are diverse and use diverse composition of primitives. Always
include atleast a few tasks which uses multi_hop_reasoning, repeat, and add primitives.
2. Task instructions should be natural user requests someone might actually ask a capable UI agent
to complete on the app. Tasks should be motivated by daily use cases.
3. For any instruction if there are going to be templated parameters then add it in following
format: {param_name}
4. Ensure that instructions are unambiguous and clearly describes an actual task that can be
performed.
5. Create realistic, executable tasks that combine multiple primitives logically in a meaningful
sequence.
6. Include primitive composition as a function-like string (e.g., "search(product,
filter=[filter(price, 50), filter(location, NYC)])")
7. For all tasks that require creating/adding/editing/deleting any information/entity make sure
all required entities for completing the task are parameterized/templated. For example:
   a.) For tasks that require adding items, both the item details and quantities should be
   specified as parameters.
   b.) For tasks that require filtering, the filter criteria should be templated to allow diverse
   task variants.
   c.) For tasks requiring multi-hop reasoning, the reasoning criteria should be clearly
   parameterized.
8. IMPORTANT: Do not include tasks that require accessing/uploading/capturing content from real
world. For example, scanning documents, recording videos, taking pictures using camera.
9. IMPORTANT: Do not generate tasks which require login.
10. Tasks should demonstrate composition of primitives working together, not just sequential
execution of unrelated actions.
11. Include template parameters where appropriate for task variation, with at least 5 possible
values for each parameter.
12. Use the provided image context to ground the task in the screenshots. Don't hallucinate any
template parameters.
```

---

## C.2 EXPLORATION AGENT

Our explorer agent is instantiated using the implementation of the task executor agent mentioned in Appendix C.3 with two key differences. First, the task instruction for each exploration turn is a generic exploration goal of the format "Explore the APP_NAME app exhaustively to access all features, functionalities and data stored on the app.". In addition, across multiple runs of exploration the explorer agent maintains an explicit memory of environment state from the past exploration turns in the same application. This memory is a text representation and it is only kept in context

when running multiple exploration turns for the same application. In order to convert a exploration trajectory to the memory representation we use GPT-4o OpenAI et al. (2024) as our summarizer MLLM. Specifically, we give the sequence of observations from the exploration trajectory as input to the summarizer MLLM and ask it to output a structured response that describes the functionalities of the environment observed by they agent in the exploration turn followed by a description of data or configs stored in the application that would be relevant for task curation. The prompt used for the summarizer MLLM is described in Appendix C.2.

---

**Summarizer MLLM Prompt (Mobile-Use: Android and Computer-Use: Ubuntu)**

```
You are a capable UI understanding agent. You will be provided a sequence of images from an app or
a website that shows how to access different features and the data currently stored in the app.
Your task is to summarize the actions taken and features, functionalities navigated and interacted
with by the user in detail.

Your task is to output the information in following format:
{

    "action summary": "A bulleted list of actions taken and pages visited to explore the features
    and functionalities explored by the agent that can enable generation of meaningful UI control
    tasks. Describe the pages and features visited in the order they were explored."

    "data stored": "A bulleted list of data found on the explored by the agent that can enable
    generation of meaningful UI control tasks. Describe the data uncovered in the order of the
    features visited as they were explored."
}
```

---

### C.3 TASK EXECUTOR AGENT

Our task executor agent is instantiated as a modular agent that decomposes the task execution into high-level task planning using a high-level planner MLLM and low-level action execution using grounding model. In addition, the high-level planner uses a reflection tool which takes as input the past observation, past action taken, and current observation to describe the transition caused by the action. The high-level planner MLLM takes as input the task instruction, current observation, history of past actions, reflection trace of past action, and outputs a high-level action. If the high-level action are click-based interaction then it uses the grounding model to localize the coordinate location.

For mobile domains, we use GPT-4o (OpenAI et al., 2024) as both the planner and reflection model, and UI-TARS-1.5 7B (Qin et al., 2025) as the grounding model. We provide the prompt used by the high-level planner in Appendix C.3, the reflection tool prompt in Appendix C.3, and the prompt for UI-TARS-1.5 7B grounding model in Appendix C.3.

For desktop domains, we use GPT-4o as the planner and reflection model, and GTA1-7B (Yang et al., 2025) as the grounding model. We provide the prompt used by the high-level planner in Appendix C.3, the reflection tool prompt in Appendix C.3, and the prompt for GTA1-7B grounding model in Appendix C.3. In the desktop setting, we additionally supply the high-level policy with heuristic actions (detailed in Appendix F) to improve data collection policy success rate.

---

**High-Level Planner MLLM Prompt (Mobile-Use: Android)**

```
You are a capable GUI assistant designed to help users navigate and interact with mobile
applications. At the beginning of each task, you will be provided with a natural language
description of the task.

Then, at each step, you will be provided with:
1. the screen image.
2. the history of actions taken on the environment and any feedbacks from an expert critic
3. feedback on the last action taken.

Your task is to analyze the goal, current screen, history of actions and feedback about the last
action, think step-by-step and generate a natural language plan that clearly describes the next
action with relevant details element or area on the screen for interaction. Finally, also output
the next action described in natural language a single sentence. Only plan for next action using
actions described above. Drag action is not supported, use clicks instead of drag.

Here are some more guidelines:
1. If the task has been completed, you should call the terminate action.
```

---

```
2. Avoid getting stuck in trying to call the same action over and over again. If you think you are
stuck, try to find alternative ways to complete the task.
3. Try to accomplish the task with the least number of actions.
4. If the previous action failed, take corrective action by taking an alternative action.

Goal: {TASK_INSTRUCTION}

Action History:
{ACTION_HISTORY}

Critic Response for Last Action:
{REFLECTION_LLM_RESPONSE}

Summary of screen changes:
{TRANSITION_SUMMARY}

Instructions: Based on the goal, current screen, history of actions containing feedback about the
last action, think step-by-step and provide the plan and next action.

At each timestep output the next action in following format:
{
    reason: "Step-by-step thinking for the action to be taken.",

    action: "The action to be taken. Choose one of the available actions. tap_on_element if you
    want to tap on an element, long_press_on_element if you want to long press on a
    element/location, type_text_in_element if you want to type text in an element, scroll_screen
    if you want to scroll the screen, answer if you want to answer the question asked by the user,
    terminate if you want to terminate the session, navigate_back if you want to navigate to
    previous screen, navigate_home if you want to navigate to the home screen, wait if you want to
    wait for 5 seconds before continuing, open_app if you want to open a specific app.",

    element_description: "A descriptions containing visual and textual details as well as spatial
    location of the UI element to be interacted with. Only set the element description if the
    action is tap_on_element, long_press_on_element, or type_text_in_element, otherwise set the
    element description to be an empty string. Examples of descriptions: circular design with a
    pattern resembling a camera shutter or a mandala, in white on a light brown background.
    Otherwise set the element description to be an empty string. Select the checkbox next to the
    label 'Remember me'.",

    text: "The text to be entered in the UI element. Only set with value to type in the text
    element.",

    direction: "The direction to scroll the screen. Can be one of [UP, DOWN, LEFT, RIGHT].",

    answer: "The answer to the question asked by the user. Only set this field if the action is
    answer, otherwise set the answer to be an empty string.",

    app_name: "The name of the app to be opened. Only set this field if the action is open_app,
    otherwise set the app_name to be an empty string."
}

Current Observation:
{OBSERVATION}
```

### Reflection MLLM Prompt (Mobile-Use: Android and Computer-Use: Ubuntu)

```
You are an expert in interacting and navigating mobile applications. Your task is to provide
useful feedback for a human to achieve a provided goal.

You will be given:
1. the screen image with detected elements before the human takes an action.
2. the description of the action the human takes along with any reason and references to detected
elements before.
3. the screen image with detected elements after the human takes an action.

Your task is to think carefully and analyze the current action, as well as the screen before and
after the action. Use this information to describe the changes in the screen and provide feedback
about whether
the action was successful or not.
If the action changes state on the screen but does not open a new view, focus on the element which
was effected to assess success.
If the action was a scroll or swipe and the UI elements did not change, the action likely failed.
If the action was to type in the text field, the evaluation should be whether or not the textfield
has the exact text typed in.
Don't tell the human what to do, just provide feedback on whether the action was successful or not.
```

```
Goal: {TASK_INSTRUCTION}

Observation before action:
{PREVIOUS_OBSERVATION}

Action executed:
{ACTION}

Observation after action:
{CURRENT_OBSERVATION}
```

### Low-Level Planner MLLM Prompt - UI-TARS 1.5 7B (Mobile-Use: Android)

```
You are a helpful assistant.

You are a GUI agent. You are given a task and your action history, with screenshots. You need to
perform the next action to complete the task.
## Output Format
```
Thought: ...
Action: ...
```
## Action Space

click(point='<point>x1 y1</point>')
long_press(point='<point>x1 y1</point>')
type(content='') #If you want to submit your input, use "\\n" at the end of `content`.
scroll(point='<point>x1 y1</point>', direction='down or up or right or left')
open_app(app_name=\'\')
drag(start_point='<point>x1 y1</point>', end_point='<point>x2 y2</point>')
press_home()
press_back()
finished(content='xxx') # Use escape characters \\', \\", and \\n in content part to ensure we can
parse the content in normal python string format.

## Note
- Use English in `Thought` part.
- Write a small plan and finally summarize your next action (with its target element) in one
sentence in `Thought` part.

## User Instruction
{TASK_INSTRUCTION}

{CURRENT_OBSERVATION}
```

### High-Level Planner MLLM Prompt (Computer-Use: Ubuntu)

```
You are an agent which follow my instruction and perform desktop computer tasks as instructed.
You have good knowledge of computer and good internet connection and assume your code will run on
a computer for controlling the mouse and keyboard.
You are on Ubuntu operating system and the resolution of the screen is 1920x1080.
For each step, you will get:
- An observation of an image, which is the screenshot of the computer screen and you will predict
the action of the computer based on the image.
- Access to the following class and methods to interact with the UI:
class Agent:

    def click(self, instruction: str, num_clicks: int = 1, button_type: str = 'left', hold_keys:
    List = []):
    '''Click on the element
        Args:
            instruction:str, decribe the element you want to interact with in detail including the
            visual description and function description. And make it clear and concise. For
            example you can describe what the element looks like, and what will be the expected
            result when you interact with it.
            num_clicks:int, number of times to click the element
            button_type:str, which mouse button to press can be "left", "middle", or "right"
            hold_keys:List, list of keys to hold while clicking
        '''

    def done(self, return_value: Union[Dict, str, List, Tuple, int, float, bool, NoneType] = None):
    '''End the current task with a success and the required return value'''
```

```
    def drag_and_drop(self, starting_description: str, ending_description: str, hold_keys: List =
    []):
    '''Drag from the starting description to the ending description
        Args:
            starting_description:str, a very detailed description of where to start the drag
            action. This description should be at least a full sentence. And make it clear and
            concise.
            ending_description:str, a very detailed description of where to end the drag action.
            This description should be at least a full sentence. And make it clear and concise.
            hold_keys:List list of keys to hold while dragging
        '''

    def fail(self):
    '''End the current task with a failure, and replan the whole task.'''

    def highlight_text_span(self, starting_phrase: str, ending_phrase: str):
    '''Highlight a text span between a provided starting phrase and ending phrase. Use this to
    highlight words, lines, and paragraphs.
        Args:
            starting_phrase:str, the phrase that denotes the start of the text span you want to
            highlight. If you only want to highlight one word, just pass in that single word.
            ending_phrase:str, the phrase that denotes the end of the text span you want to
            highlight. If you only want to highlight one word, just pass in that single word.
        '''

    def hold_and_press(self, hold_keys: List, press_keys: List):
    '''Hold a list of keys and press a list of keys
        Args:
            hold_keys:List, list of keys to hold
            press_keys:List, list of keys to press in a sequence
        '''

    def hotkey(self, keys: List):
    '''Press a hotkey combination
        Args:
            keys:List the keys to press in combination in a list format (e.g. ['ctrl', 'c'])
        '''

    def open(self, app_or_filename: str):
    '''Open any application or file with name app_or_filename. Use this action to open
    applications or files on the desktop, do not open manually.
        Args:
            app_or_filename:str, the name of the application or filename to open
        '''

    def scroll(self, instruction: str, clicks: int, shift: bool = False):
    '''Scroll the element in the specified direction
        Args:
            instruction:str, a very detailed description of which element to enter scroll in. This
            description should be at least a full sentence. And make it clear and concise.
            clicks:int, the number of clicks to scroll can be positive (up) or negative (down).
            shift:bool, whether to use shift+scroll for horizontal scrolling
        '''

    def set_cell_values(self, cell_values: Dict[str, Any], app_name: str, sheet_name: str):
    '''Use this to set individual cell values in a spreadsheet. For example, setting A2 to "hello"
    would be done by passing {{"A2": "hello"}} as cell_values. The sheet must be opened before
    this command can be used.
        Args:
            cell_values: Dict[str, Any], A dictionary of cell values to set in the spreadsheet.
            The keys are the cell coordinates in the format "A1", "B2", etc.
                Supported value types include: float, int, string, bool, formulas.
            app_name: str, The name of the spreadsheet application. For example, "Some_sheet.xlsx".
            sheet_name: str, The name of the sheet in the spreadsheet. For example, "Sheet1".
        '''

    def switch_applications(self, app_code):
    '''Switch to a different application that is already open
        Args:
            app_code:str the code name of the application to switch to from the provided list of
            open applications
        '''

    def type(self, element_description: Optional[str] = None, text: str = '', overwrite: bool =
    False, enter: bool = False):
    '''Type text into a specific element
        Args:
            element_description:str, a detailed description of which element to enter text in.
            This description should be at least a full sentence.
```

```
            text:str, the text to type
            overwrite:bool, Assign it to True if the text should overwrite the existing text,
            otherwise assign it to False. Using this argument clears all text in an element.
            enter:bool, Assign it to True if the enter key should be pressed after typing the
            text, otherwise assign it to False.
        '''

    def wait(self, time: float):
    '''Wait for a specified amount of time
        Args:
            time:float the amount of time to wait in seconds
        '''

The following rules are IMPORTANT:
- If previous actions didn't achieve the expected result, do not repeat them, especially the last
one. Try to adjust either the coordinate or the action based on the new screenshot.
- Do not predict multiple clicks at once. Base each action on the current screenshot; do not
predict actions for elements or events not yet visible in the screenshot.
- You cannot complete the task by outputting text content in your response. You must use mouse and
keyboard to interact with the computer. Call ```agent.fail()``` function when you think the task
can not be done.
- You must use only the available methods provided above to interact with the UI, do not invent
new methods.

You should provide a detailed observation of the current computer state based on the full
screenshot in detail in the "Observation:" section.
Provide any information that is possibly relevant to achieving the task goal and any elements that
may affect the task execution, such as pop-ups, notifications, error messages, loading states,
etc..
You MUST return the observation before the thought.

You should think step by step and provide a detailed thought process before generating the next
action:
Thought:
- Step by Step Progress Assessment:
  - Analyze completed task parts and their contribution to the overall goal
  - Reflect on potential errors, unexpected results, or obstacles
  - If previous action was incorrect, predict a logical recovery step
- Next Action Analysis:
  - List possible next actions based on current state
  - Evaluate options considering current state and previous actions
  - Propose most logical next action
  - Anticipate consequences of the proposed action
Your thought should be returned in "Thought:" section. You MUST return the thought before the code.

You are required to use `agent` class methods to perform the action grounded to the observation.
Return exactly ONE line of python code to perform the action each time. At each step (example:
```agent.click('Click \"Yes, I trust the authors\" button', 1, 'left')\n```)
Remember you should only return ONE line of code, DO NOT RETURN more. You should return the code
inside a code block, like this:
```python
agent.click('Click \"Yes, I trust the authors\" button', 1, "left")
```

For your reference, you have maximum of {MAX_STEPS} steps, and current step is {CURRENT_STEP} out
of {MAX_STEPS}.
If you are in the last step, you should return ```agent.done()``` or ```agent.fail()``` according
to the result.

Here are some guidelines for you:
1. Remember to generate the corresponding instruction to the code before a # in a comment and only
return ONE line of code.
2. `agent.click` can have multiple clicks. For example, agent.click('Click \"Yes, I trust the
authors\" button', 2, "left") is double click.
3. Return ```agent.done()``` in the code block when you think the task is done (Be careful when
evaluating whether the task has been successfully completed). Return ```agent.fail()``` in the
code block when you think the task can not be done.
4. Whenever possible, your grounded action should use hot-keys with the agent.hotkey() action
instead of clicking or dragging.
5. Save modified files before returning ```agent.done()```. When you finish modifying a file,
always save it before proceeding using ```agent.hotkey(['ctrl', 's'])``` or equivalent. Tasks may
involve multiple files. Save each after finishing modification.
6. If you meet "Authentication required" prompt, you can continue to click "Cancel" to close it.

My computer's password is 'password', feel free to use it when you need sudo rights.

Task Instruction:
{TASK_INSTRUCTION}
```

```
Current Observation:
{OBSERVATION}
```

---

**Low-Level Planner MLLM Prompt - GTA1-7B (Computer-Use: Ubuntu)**

```
You are an expert UI element locator. Given a GUI image and a user's element description, provide
the coordinates of the specified element as a single (x,y) point. The image resolution is height
{HEIGHT} and width {WIDTH}. For elements with area, return the center point.

Output the coordinate pair exactly:
(x,y)

Task Instruction: {TASK_INSTRUCTION}
{CURRENT_OBSERVATION}
```

## C.4 TASK VERIFIER

To verify if a task executed by the data collection policy is executed successfully, we employ a MLLM as a task verifier that evaluates trajectories. The verifier is an MLLM that takes as input the task instruction and the executed trajectory (represented as interleaved images and actions). For each example it outputs the following in order: (i) screen_details: summarizing what the agent is doing in the trajectory, (ii) reasoning: producing a Chain-of-Thought (Wei et al., 2023) justification of whether the instruction has been completed, and (iii) result: issuing a final judgment of "success" or "failure." We use GPT-4o (OpenAI et al., 2024) as the task verifier on expert-collected trajectories and Qwen2.5-VL 32B Instruct ](Bai et al., 2025) for RL training. The prompt used by both verifier models is specified in Appendix C.4.

---

**Task Verifier Prompt (Mobile-Use: Android and Computer-Use: Ubuntu)**

```
You are an AI assitant designed to help users evaluate whether a specific interaction with mobile
application was successful or not. At the begining of the task you will be provided:

1. a description of the task
2. a sequence of UI screenshots interleaved with actions taken in order to complete the specified
task

For each action taken, you will be provided the action type, location shown on the screen with red
dot indicating where the interaction executed, and reason for the action taken.

Your task is to carefully look at all the screenshots shown, the description of the task, and
actions taken to evaluate whether the specified task was completed or not. A task is only
considered completed if the screenshots only show the intended task being achieved. If the screens
show any unintended changes to the state of the app then the demonstration of task is treated as a
failure.

Task instruction: {TASK_INSTRUCTION}

Observations: {OBSERVATIONS_WITH_ACTIONS}

You should output the evaluation response in the following JSON format:
{
  "screen_details": "A bulleted list of all the changes on the screens as a result of all actions
  executed. Even if the screen change do not correspond to the intended actions, you should still
  describe the screen changes. Create a detailed list of all the UI changes that occurred on the
  screen as a result of all the actions."

  "reasoning": "Step-by-step reasoning for the assesment of whether the agent was successful for
  the current step or if the agent has failed for the current step.",

  "result": "Assessment of the task completion given the last screen. Choose one of the available
  assessment result.success if the agent has successfully completed the assigned task, fail if you
  think the agent was unsuccessful in performing the given task"
}
```

## D TASK CATEGORIZATION FOR ANALYSIS

To support the analysis in Sec. 4.2, each task is classified to multiple task categories that describe the types of skills required by an agent to complete the task. To annotate each task, we use an LLM

| | Task Category | Description |
|---|---|---|
| 1) | create | creates a new entity in the app. |
| 2) | edit | modifies an existing entity in the app. |
| 3) | delete | deletes an entity from the app. |
| 4) | search | searches for something using the search bar. You can optionally filter the results based on the given criteria. |
| 5) | filter | filters the results based on the given criteria. You can optionally compose multiple filters to form a single filter. A filter can be direct, or it needs to be inferred using multi hop reasoning. |
| 6) | repeat operation | repeatedly applies a certain create/edit/delete/search operation multiple times to execute a task. For example, deleting multiple events in calendar or all events on a single day. |
| 7) | information retrieval | retrieves information from the app. This can involve answering questions about certain setting or data from an app |
| 8) | multi hop reasoning | involves reasoning across multiple steps or pieces of information to arrive at a conclusion. For example, answering a question that requires searching for data using multi-hop reasoning. |
| 9) | edit system setting | modifies a setting at the system level on the device. |
| 10) | edit app setting | modifies a setting within a specific application. |
| 11) | fill form | completes a form with the necessary information. |
| 12) | text editing | makes changes to text, such as editing, formatting, or restructuring. |
| 13) | cross app interaction | involves interacting with multiple applications to complete a task. |
| 14) | complex ui interaction | involves interacting with complex UI elements like timers and date time picker wheels. |
| 15) | composition | involves executing a task that requires completing multiple sub-tasks of different task categories to complete a longer horizon task. For example, tasks that require adding a new contact and deleting one involves executing both create and delete operation. |

Table 7. **Task Categories** used for categorization of mobile-use tasks on Android and AndroidWorld.

(GPT-4o (OpenAI et al., 2024) in this case) and prompt it using the prompt specified in Appendix D using platform-specific categories specified in Tab. 7 for mobile-use agent tasks and in Tab. 8 for computer-use agent tasks.

---

**Task Classifier Prompt (Mobile-Use: Android and Computer-Use: Ubuntu)**

```
You are a helpful assistant that can help me analyze the task can identify the task categories
which uniquely describe the types of tasks and skills required by a GUI agent to complete the
task. As input you will be provided a list of task instructions and your task is to output the
list of task categories that apply to the specified instructions. Next, we will list the type of
task categories you need to label each task using following categories:

{TASK_CATEGORIES}

For each task instruction, you have to output a response in JSON format.
- A list of task categories that apply to the specific task instruction.
- Use only the list of task categories listed above as part of the response.

Here are some examples of task categorization:
Instructions: [
"Create a timer with {hours} hours, {minutes} minutes, and {seconds} seconds. Do not start the
timer.",
"Add the expenses from expenses.jpg in Simple Gallery Pro to pro expense.",
"Change the theme to {theme} in the Audio Recorder app."
]
Output:

[
    {
        "task_instruction": "Create a timer with {hours} hours, {minutes} minutes, and {seconds}
        seconds. Do not start the timer.",
        "task_categories": [
```

| | Task Category | Description |
|---|---|---|
| 1) | create | creates a new entity in the application. |
| 2) | edit | modifies an existing entity in the application. |
| 3) | delete | deletes an entity from the application. |
| 4) | filter | filters search results or data to operate on based on the given criteria. You can optionally compose multiple filters to form a single filter. A filter can be direct, or it needs to be inferred using multi hop reasoning. |
| 5) | repeat operation | repeatedly applies a certain create/edit/delete/search operation multiple times to execute a task. For example, deleting multiple events in calendar or all events on a single day. |
| 6) | information retrieval | retrieves information from the app. This can involve answering questions about certain setting or data from an app |
| 7) | multi hop reasoning | involves reasoning across multiple steps or pieces of information to arrive at a conclusion. For example, answering a question that requires searching for data using multi hop reasoning. |
| 8) | file management | involves managing files and folders on the computer. |
| 9) | coding | involves writing code to accomplish a specific task or solve a problem. |
| 10) | web search | searches for information on the web using a search engine. |
| 11) | device search | searches for information on the computer file system. |
| 12) | cross app tasks | involves interacting with desktop multiple applications to complete a task. |
| 13) | edit app setting | modifies a setting within a specific application. |
| 14) | edit system settings | modifies a setting at the system level on the computer. |
| 15) | composition | involves executing a task that requires completing multiple sub tasks of different task categories to complete a longer horizon task. For example, tasks that require adding a new contact and deleting one involves executing both create and delete operation. |
| 16) | content editing | tasks that require editing content such as text, images, videos, or audio. |
| 17) | spreadsheet editing | tasks that require editing or manipulating data in a spreadsheet application. |
| 18) | document editing | tasks that require editing or formatting documents, such as word processing files or PDFs. |
| 19) | bash scripting | tasks that require writing or executing bash scripts or commands using the terminal or code editors. |

Table 8. **Task Categories** used for categorization of computer-use tasks on Ubuntu and OSWorld.

```
                "create",
                "complex ui interaction"
            ]
        },
        {
            "task_instruction": "Add the expenses from expenses.jpg in Simple Gallery Pro to pro
            expense.",
            "task_categories": [
                "create",
                "cross app interaction",
                "fill form"
            ]
        },
        {
            "task_instruction": "Change the theme to {theme} in the Audio Recorder app.",
            "task_categories": [
                "edit app setting"
            ]
        }
    }
]

Next, you will be provided 10 tasks as input and your task is to output a JSON in specified format:
```

## E  TASK PROPOSER ANALYSIS

In this section, we describe the prompts and setup used to synthesize data used for *No Exploration* and *Iterative Exploration* task generators for comparison with AUTOPLAY.

***No Exploration***: We use the prompt mentioned in Appendix E to instantiate the *No Exploration* method. In order to provide high-quality environment context we manually write text descriptions for all 20 android applications we use for generating training data. Each of these description aims to enumerate the list of functionalities accessible in an application. This information is gathered by us manually by exploring each android application manually which is a key limitation of this method. We show example descriptions of two applications in Appendix E.

***Iterative Exploration***: To instantiate the *Iterative Exploration* method we use the public implementation of AgentSynth (Xie et al., 2025) as it demonstrates a effective pipeline for iterative exploration for data collection for web agents. We adapt AgentSynth (Xie et al., 2025) for mobile-use domain by using the data collection policy described in Sec. 3.3 as the task executor. We run the iterative task proposal and execution for $k$ turns where $k$ is randomly sampled to be between $3 - 8$ turns and each turn can run up to 7 steps. We use 7 steps as a limit for each subtask as we qualitatively find that subtasks proposed by such methods can be completed with $4 - 7$ steps on average. After all $k$ turns are executed the complete trajectory is hindsight relabelled using descriptions of each subtask and whether they were successfully completed or not. Using this setup, we run this method to generate $5k$ tasks and use trajectories where atleast $50\%$ subtasks succeeded as our training dataset.

---

**No Exploration Task Generator MLLM Prompt**

```
You are a capable UI understanding agent. You will be provided a app description for the Android
app. Your task is to propose a list of tasks that a useful UI understanding agent should be able
to complete. Use the app description provided to ground these tasks to the ones that are feasible.
Propose as many diverse tasks as possible to cover broad range of features.

App description: {app_description}

For all described functionalities output a list of up to {NUM_TASKS} unique tasks that can be
executed in the app as a JSON which should be in the following format:

tasks = [
    {
        "thought": "<Detailed thought and reasoning for the proposed task, why is the task simple
        enough to execute and whether it satisfies the task proposal guidelines>",
        "instruction": <natural language instruction describing a task with/without template
        params with name of the app>,
        "tag": <few words describing the type of task>,
        "app_name": <name of the app>,
        "template params": {
            "param_name": {
                "description": <param description>,
```
```

```
                    "possible_values": [<list of 5 random values>]
                }
            }
        }
    }
]

For each functionality/feature make sure to follow these guidelines while generating tasks:
1. Create all possible tasks. For example, if there is a clock app on the phone which has
worldwide clock feature then you can create tasks like: "Add worldwide clock for <city>", "Remove
worldwide clock for <city>", "Add worldwide clock for <city_1, city_2> and reorder to put <city_2>
before <city_1>". Make sure to add the <param> details in "template params" in JSON to allow
creation of diverse tasks.
2. In any instruction if there are going to be templated parameters then add it in following
format: {param_name}
3. If a task requires creating/adding/editing/deleting any information/event/entry make sure all
required entities for completing the task are parameterized/templated. For example:
    a.) For a task that requires creating notes both the name of the new note and content of the
    note should be specified as parameters.
    b.) Similarly, if a task requires editing some details describe the exact edit you'd like to
    make, what entity needs to be edited. When proposing such tasks, only ask edits to existing
    entities shown in the images from the app for such tasks. If no such entities exist ask the
    agent to first create one and then edit it.
    c.) If a task requires deleting an entity, describe the entity that needs to be deleted. When
    proposing such a task, only ask to delete an entity that already exists as shown in the images
    from the app for such tasks. If no such entities exist ask the agent to first create one and
    then delete it.
    d.) If a task requires copying an entity, describe the entity that needs to be copied and
    where it should be copied to (ex: which folder or date). When proposing such a task, only ask
    to copy an entity that already exists as shown in the images from the app for such tasks. If
    no such entities exist ask the agent to first create one and then copy it.
    e.) For all parameters that are templated if any parameter refers to an entity for edit,
    delete or copy task make sure that the list of possible values only contain entities that
    exists or will be created.
4. Ensure that instructions are unambiguous and clearly describes a actual task that can be
performed on the app.
5. Do not include tasks that require accessing/uploading/capturing content from real world. For
example, scanning a document, recording a new video, taking a picture using camera.
6. Task instructions should be natural user requests someone might actually ask a capable UI agent
to complete on the app.
7. If any supports browsing information from the web or library make sure to include that in the
task instruction. For example, if a task requires searching for a book on the web, make sure to
include that in the task instruction.
```

## App Description Examples (Mobile Use: Android)

```
### Audio Recorder app

The Audio Recorder app is a versatile and user-friendly application designed for recording audio
with a range of customizable settings. It caters to personal, educational, and professional audio
recording needs by offering options to select recording formats, sample rates, bitrates, and
channel counts. The app also provides features for theme customization, file management, and
sharing, making it suitable for various audio capture and management tasks.

"Theme Customization: Users can personalize the app's appearance by selecting from themes such as
Blue Gray, Black, Teal, Blue, Purple, Pink, Orange, Red, and Brown. This feature enhances the user
experience by allowing visual customization.",
"Recording Format Selection: Users can choose between M4a, Wav, and 3gp formats. M4a is
recommended for its good quality and small size, Wav is uncompressed and takes more space, and 3gp
is suitable for saving space. This feature allows users to tailor the recording quality and file
size to their needs.",
"Sample Rate Selection: Offers options for 8kHz, 16kHz, 22kHz, 32kHz, 44.1kHz, and 48kHz, enabling
users to select the desired audio quality. Higher sample rates provide better audio quality.",
"Bitrate Selection: Users can choose from 48 kbps, 96 kbps, 128 kbps, 192 kbps, and 256 kbps to
control the audio quality and file size. Higher bitrates result in better audio quality.",
"Channel Count: Options for Stereo and Mono recording, providing flexibility in audio capture.
Stereo offers two channels for richer sound, while Mono uses a single channel.",
"File Management: Includes a file browser to access recorded files, options to rename, view
detailed information (format, bitrate, channel count, sample rate, duration, size, file location,
creation date), organize files by date, and a trash feature where deleted files are stored for 60
days before permanent deletion. This feature aids in efficient file organization and retrieval.",
"Recording Interface: Displays recording time, file size, format, and sample rate during
recording, with easy access to settings and file management. Users can start, pause, and stop
recordings, and view a visual waveform of the audio being recorded.",

================================================================================
================================================================================
```

| Action | Description |
|---|---|
| click | Click on (x, y) coordinate |
| long_press | Tap and hold on (x, y) coordinate |
| input_text | Types 'text' in a textbox if active |
| open_app | Opens android application given as text argument |
| scroll | Scrolls screen in direction (up or down) |
| wait | Waits for 'time' specified in seconds |
| navigate_back | Navigates back to previous screen |
| navigate_home | Navigates to home screen of the device |
| terminate | Ends the current episode |

**Table 9. Action Space** used for mobile-use agent evaluated on AndroidWorld.

```
### Simple Calendar Pro

This app is a versatile calendar application designed to help users efficiently manage their
schedules for both personal and professional use. It offers a range of features including event
and task management, multiple calendar views, import/export functionality, and customization
options. Users can create and organize events and tasks, set reminders, and customize their
calendar experience with color and notification settings. The app supports adding holidays,
birthdays, and anniversaries, and provides options for printing schedules and navigating through
dates.

"Calendar View: Displays various views such as daily, weekly, monthly, and yearly, allowing users
to navigate through months and view scheduled activities. Users can customize their view
preferences and highlight weekends.",
"Event Creation: Users can create new events with details such as title, location, description,
start and end time, reminders, repetition settings, event type, and color customization. Events
can be added, edited, and deleted.",
"Task Creation: Users can add tasks to their calendar, similar to events, to manage to-do lists
and deadlines. Tasks can be organized and viewed in different calendar views.",
"Search Functionality: Users can search for specific events or tasks using keywords, making it
easy to find events quickly.",
"Import/Export Events: Users can import events from an .ics file and export their calendar events
to an .ics file, facilitating easy sharing and backup of schedules.",
"Print Functionality: Users can print their schedules, with options to select the number of copies
and paper size. The app supports saving the schedule as a PDF or printing via connected printers.",
"Add Holidays and Contacts: Users can add holidays, contact birthdays, and anniversaries to their
calendar, ensuring they never miss important dates. The app may request access to contacts for
this feature.",
"Event Reminders: Users can set reminders for events to receive notifications. Options include
setting no reminder or adding new birthdays and anniversaries automatically.",
"Settings Customization: Offers options to customize colors, widget colors, language settings,
time format (24-hour), week start day, and weekend highlighting. Users can also customize
notifications, choose audio streams for reminders, and enable vibration for notifications.",
"Color Customization: Users can change the theme and app icon color, with a warning about
potential issues with some launchers.",
"Go to Date: Allows users to quickly navigate to a specific date in the calendar.",
"About: Provides information about the app."

...
```

# F   ACTION SPACE

For mobile-use agents, we use the low-level action space from AndroidWorld (Rawles et al., 2024) benchmark. The full list of actions used by our data collection policy and AUTOPLAY finetuned models is listed in Tab. 9.

For training desktop-use agents, we use a hybrid action space implemented in Agent-S Agashe et al. (2025) and used by prior works (Yang et al., 2025). The full list of actions used by our data collection policy and AUTOPLAY finetuned models is listed in Tab. 10. At the low-level these actions are implemented using pyautogui APIs widely used in methods on the OSWorld benchmark (Xie et al., 2024).

| Action | Description |
|---|---|
| click | Click on (x, y) coordinate, num_click times, using either left or right click button, while holding keys given by 'hold_keys' |
| hold_and_press | Holds list of keys 'hold_keys' and presses keys given by 'press_keys' |
| hotkey | Invokes a hotkey combination given by 'keys' |
| open | Opens the file name or application specified as the argument |
| scroll | Clicks on (x, y) coordinate, scrolls in up or down direction while holding shift key if 'shift' argument is set to true |
| set_cell_values | Sets individual cells in a spreadsheet to certain values |
| switch_applications | Switch to a different application that is already open specified as argument |
| type | Type specified text in an element |
| wait | Wait for a specified amount of time in seconds |
| fail | Ends the current episode and returns failure response |
| done | Ends the current episode |

**Table 10. Action Space** used for computer-use agent evaluated on OSWorld. We use pyautogui action space to implement the specified actions.

## G USE OF LLM FOR WRITING

We use LLMs to assist with specific aspects of paper writing which includes: fix grammatical errors, sentence polishing, and paraphrasing parts of for sections 1, 3 and 4. LLMs were used for: (1) grammar checking and improving clarity and readability of text, (2) suggestions for writing certain sentences via paraphrasing. For each section the original content was written by the authors and edited using GPT-4o OpenAI et al. (2024) LLM. Followed by this round of LLM-assisted editing, the authors verified and polished the content written by the LLM to fix issues with generated text and create the final text used for the paper.

