# OpenReview forum: "Scaling Synthetic Task Generation for Agents via Exploration"
_ICLR.cc/2026/Conference — ICLR 2026 Poster_

### Official Review · Reviewer_Pkt3 · 2025-10-20

**Soundness:** 2
**Presentation:** 1
**Contribution:** 1
**Rating:** 2
**Confidence:** 3

**Summary:**

The paper implements a system where an LLM explores a (simulated) environment to generate exploration traces and these traces are hindsight relabeled by an LLM prompted to define types of tasks to provide with goals for those traces. The method is applied in the domain of graphical user interface interactions, evaluated in the Android World and OSWorld environments. The exploration process includes compression of past experiences into text summaries for use in memory to support diversity of outputs while addressing context length limitations of LLMs. Traces from the system are used for supervised finetuning of VLMs and can be combined with the generated tasks to support VLM verifiers for RL finetuning.

Evaluations compare the model against baseline UI agents and ablations of the model.

**Strengths:**

# originality
Primarily in the form of bringing ideas for automated exploration to bear on the GUI agent domain.

# quality
Evaluates against strong baselines and in two different, relevant domains.

# significance
Scalable data generation for VLM agent training is broadly relevant to agentic AI systems in general: whether digital agents (GUI-based, here) or physical agents (robots or game playing).

**Weaknesses:**

# originality
Few elements of the proposal are unique on their own. The novelty is tied to idea combination.

# quality
It is unclear how much the model improves compared to baselines, and by how much or under which conditions. See below for detailed questions. Currently it is not yet clear to me that the framework can do more than modestly improve a weak model, which sometimes is outperformed by a baseline that ablates the exploration method (at least in some scenarios).

The results seem quite mixed at the moment and the scaling and costs are not clearly addressed.


# clarity
It is very difficult to follow the experiment results narrative and how it relates to the tables. Concrete points are below. Generally the definitions and abbreviations are not clear and not well-connected by the text or captions. Figure 3 is particularly cumbersome to understand.


# significance
The approach is tailored to the specific GUI task domains through the task guideline prompts and perhaps a few other design decisions. This will modestly limit the scope to the GUI agent audience.

**Questions:**

# questions
- How do agent performance results scale with the generated dataset size?
	- Even with the existing dataset, it would help to show how downstream agent training improves with subsets of the total generated interactions. This could help strengthen the claims that this approach is valuable by showing that it can provide strongly improving baseline model improvements. Right now the results saturate below strong baselines.
- How do these results scale with repetition of the whole loop? That is: explore, train the agent (SFT & RL), then explore again.
	- This would strengthen the claim that the method is scalable over human annotation.
- How do the costs of AutoPlay scale?
	- In terms of quantities like number of LLM calls and wall-clock compute time.
	- Also in terms of number of interactions or diversity of tasks.
	- This can strengthen the scalability claims.
- Table 1: Is there any statistical test for differences and effect sizes that could be applied to these results?


# suggestions / smaller details
- Table 2: Where are the data referenced in lines 416-420? I'm not sure I follow which data are referenced by the claims about the same executor feasibility vs agent training.
	- Why does No Exploration show such a high Pass@5 in Android World?
	- What does the high Pass@1 for Task Execution of Iterative Exploration indicate (56.4% vs Autoplay with 46.0%)?
	- I think my confusion is different terms are used in the text and the tables, with the definitions somewhere else in the text, making the results unclear. After some reviewing I believe: task execution references the fixed executor, Pass@1 means 'average success rate' (per line 344), but the 9.4% number is the percentage point (not percentage) difference between AutoPlay and No Exploration in Android World in Pass@5 (defined to be percentage of tasks where any trial is successful on lines 346-347).
- Section 3.4: Why use a different verifier (Qwen2.5-VL-32B-Instruct; lines 307-308) for RL instead of training data generation and verification (GPT-4o; lines 275-276)?
- Figure 3: I do not understand how to read this figure.
	- What are Left vs Right in the caption? The figure has 3 subfigures. (b) is labeled "AndroidWorld Tasks", but the legend says OSWorld for green.
	- Please tell me if this is the correct interpretation: dark colors are execution success rates, light colors are proportions among the total dataset. (a) shows AndroidWorld, (b) shows OSWorld, (c) is a comparison of AutoPlay to NoExploration.
	- I think this should be split into different figures that are clearer. The distribution would make sense as a pie chart or side-by-side bars or stacked bars. The success rates should be visulized separately as they are semantically different. Same for the ablation (ideally adding Iterative Exploration).
- Table 1 might benefit from including a column indicating model size. Right now many results appear underwhelming as AutoPlay performs worse than many of the baselines.

---

> ### Author Response · Authors · 2025-11-25
> **Response to reviewer**
>
> Thanks for the constructive feedback!
>
> > Q. How does agent performance scale with dataset size?
>
> A. Due to limited compute resources during the rebuttal timeline we do not conduct this experiment.
>
> > Q. How do these results scale with repetition of the whole loop? That is: explore, train the agent (SFT and RL), then explore again.
>
> A. This experiment is out of the scope of our proposed work. In our method, the policy trained with SFT and RL does not provide any feedback to the task generator that would change the task sampling distribution. We argue naively repeating this loop would not lead to meaningful improvements without an explicit feedback loop to the task generator to iteratively improve the quality of the dataset. Therefore, we leave studying these improvements as part of the future work.
>
> > Q. How do the costs of AutoPlay scale?
>
> A. Each task demonstration costs approximately $0.50 to generate, with trajectories of up to 15 steps. This estimate is obtained by counting input and output tokens used by both the task-planner LLM and the reflection-tool LLM at every step, then averaging token usage over 100 trajectories. At each step, the task planner processes the current observation image together with the text history of previous observations. The reflection LLM processes the current and past observations along with the text action. Due to this, each step includes high-resolution images (2400×1080) which require ~1532 tokens and growing text histories over trajectories up to 15 steps; leading to a single trajectory consumes roughly 120k-200k tokens.
>
> > Q. Is there any statistical test for differences and effect sizes that could be applied to these results?
>
> A. Thanks! We conduct a t-test and report standard error, p-value, and t-statistic of our SFT finetuned model in comparison with baseline models. We find all of our models show statistically significant improvements as p-value < 0.05 and |t-statistic| > 2.
>
> | Model              | Pass@1 | p-value | t-statistic |
> |-----------------------|--------|--------|--------|
> | Qwen2.5-VL Instruct 7B                | 19.5 &plusmn; 2.9  | -   | - |
> | Qwen2.5-VL Instruct 72B              | 35.0 &plusmn; 1.4  | -  | - |
> | AutoPlay 7B                | 40.1 &plusmn; 3.3  | 0.001   | 4.7 |
> | AutoPlay 72B                | 47.9 &plusmn; 1.6  | 0.0003   | 6.1 |
> | AutoPlay 3B                | 34.2 &plusmn; 1.6  | -   | - |
> | AutoPlay-RL 3B                | 39.9 &plusmn; 1.7  | 0.04   | -2.44 |
>
> > Q. Where are the data referenced in lines 416-420?
>
> A. The data referenced is the Pass@1 performance of evaluating policy training using data collected with No Exploration and Iterative Exploration baselines on AndroidWorld benchmark. These numbers are percentage point improvements of AutoPlay over the baselines presented in Table. 2. We updated the text to clarify this now.
>
> > Q. Why does No Exploration show such a high Pass@5 in Android World?
>
> A. When we look at task performance by app for Pass@5 for No Exploration policy we find that it achieves higher success rates than Pass@1 for apps which have easier tasks like Simple Calendar Pro, Pro Expense, Broccoli-Recipe App, Settings app, and SMS app.
>
> > Q. What does the high Pass@1 for Task Execution of Iterative Exploration indicate (56.4% vs Autoplay with 46.0%)?
>
> A. The Task Execution Pass@1 scores reported in Table 2 reflect how well the task executor performs on tasks generated by each of the task-generation methods. These results indicate, on average the tasks generated by Iterative Exploration achieve 56.4% task execution success rate vs. 46.0% for tasks generated using each task generator. We believe as the Iterative Exploration method proposes short horizon subtasks and chains them to create the final long-horizon task it implicitly leads to higher task execution success rate as the task executor needs to solve short horizon easier subtasks only.
>
> > Q. Why use a different verifier for RL instead of training data generation and verification?
>
> A. We use Qwen2.5-VL Instruct 32B as our verifier for RL training because it is more scalable and cheaper to run. This leads to higher training throughput and no additional API costs. In contrast, using GPT4o would incur API costs and would be bottlenecked by HTTP API throughput.
>
> > Q. Figure 3 is unclear and hard to interpret
>
> A. Thanks for the feedback. We have updated Fig. 3 in the main paper as per the suggestion.
>
> We are happy to answer any follow-ups or concerns reviewer has. If they think our response has addressed their concerns we'd appreciate if they could acknowledge it and adjust the rating for our paper.

---

> > ### Comment · Reviewer_Pkt3 · 2025-11-28
> >
> > Thank you for the responses and clarifications!
> >
> > > RE: Statistical Tests
> >
> > Great to see.
> >
> > > Q. How do the costs of AutoPlay scale?
> >
> > I was asking to clarify how much the costs would increase ask the scale of problem is changed. Maybe another way to phrase this is: how many calls are needed for a task and where do they come from? The token analysis is useful to see as well, but I was trying to understand scaling behavior for the approach overall.

---

> ### Author Response · Authors · 2025-11-28
> **Response to reviewer**
>
> > Q. I was asking to clarify how much the costs would increase ask the scale of problem is changed. Maybe another way to phrase this is: how many calls are needed for a task and where do they come from? The token analysis is useful to see as well, but I was trying to understand scaling behavior for the approach overall.
>
> A. Task executor costs scales with the number of generated demonstrations and steps per demonstration. Generating each task demonstration of up to 15 steps costs approximately $0.50$ for the AutoPlay tasks and uses two LLM API calls for each step. The first call summarizes the observation history and uses ~4k input tokens and ~1k output tokens costing approximately $0.02$. The second call generates the action from the summary and current observation, using ~2.5k input tokens and ~500 output tokens costing approximately $0.011$. Thus the cost per step is approximately $0.031$ and the cost of a demonstration is a function of the number of steps.
>
> Exploration costs scale with the number of steps per exploration turn and number of turns per environment. We run exploration for 15 steps for each environment for 3 turns. The exploration stage uses the same executor policy mentioned above. Therefore the total cost of the exploration phase for a single environment is = 0.0465 x 3 = ~ $1.395.
>
> Task proposal costs scale with the number of generated tasks. The task generator uses one exploration trajectory and produces up to 50 tasks per turn. Each turn uses one LLM call processing ~24.4k input tokens and ~10k output tokens, costing $0.22$ per 50 tasks. Costs scale linearly with the number of generated tasks. In practice, producing ~20k tasks for AndroidWorld cost ~$53.1$, and scaling to ~100k tasks would be about ~$265.44.

---

### Official Review · Reviewer_3kvg · 2025-10-22

**Soundness:** 3
**Presentation:** 3
**Contribution:** 3
**Rating:** 6
**Confidence:** 4

**Summary:**

AUTOPLAY proposes a two-stage, exploration-first pipeline to synthesize large-scale, feasible and verifiable UI-agent tasks without human labels. Stage (i) uses a goal-agnostic MLLM explorer  to discover novel states and functionalities; stage (ii) conditions an MLLM task generator on exploration trajectories plus domain task guidelines to synthesize diverse, feasible, and verifiable tasks. Using an MLLM executor and verifier, the authors collect demonstrations (no human annotation) for SFT and train with RL via verifier rewards. On AndroidWorld and OSWorld, AUTOPLAY improves success rates over base Qwen2.5-VL models across sizes (e.g., +20.6% at 7B on AndroidWorld, +10.1% at 72B on OSWorld), competes with or surpasses several strong baselines, and shows an additional +5.7% from RL. Ablations indicate both exploration and task guidelines are important.

**Strengths:**

1. The paper offers originality and significance. The paper targets a central bottleneck, scalable, feasible, and verifiable downstream task data for post-training MLLM agents—and proposes a  closed-loop pipeline  that delivers substantial improvements, suggesting a general recipe for building capable UI agents and enabling RL at scale.

2. Results show end-to-end gains across two established benchmarks (AndroidWorld/OSWorld) and multiple model scales. Ablations carefully compare each components, supporting the design choices.

3. The paper is clear and reproducible. It decomposes the method into well-scoped components,with a consistent algorithmic framing, which facilitates understanding and reproduction.

**Weaknesses:**

1. Quality of Exploration (systematicity, coverage, and realism) need further analysis. The paper claims the task generator yields diverse tasks, but the exploration stage lacks formal analysis of how exploration is systematic and how coverage is measured. It also remains unclear whether the synthesized task distribution aligns with real-world task distributions.

2. Verifier reliability and shared-model bias. Both the executor and verifier use GPT-4o, creating a risk of style/format bias. The entire pipeline (SFT and RL rewards) hinges on an MLLM verifier, yet no human-consistency study is reported.

3. Tasks are generated on the same app suites used for final evaluation (AndroidWorld/OSWorld). Without an explicit app-level or task-level split,  it is unclear how well the trained agent generalizes to hold-out, unseen apps, and there is risk of distribution leakage.

**Questions:**

1. Please provide formal metrics for exploration quality: e.g., UI-state/transition coverage, unique control coverage. How closely do generated tasks match natural task distributions? Report distribution comparisions against AndroidWorld/OSWorld category or other human-annotated datasets

2. I recommend a rigorous evaluation of the verifier. Please report precision/recall against a small, human-audited gold set and document any calibration or abstention policy. In addition, establish multi-judge consistency by adding a second, independent judge (e.g., an open-source LLM or simple rule-based checks) and comparing accuracy and agreement; this would materially increase confidence in the generated data quality and the reliability of rewards.

3. Did you enforce strict isolation so that evaluation tasks/functions never appeared during task generation? Is there an OOD split (cross-app / cross-task )? Any results under UI perturbations (resolution, locale, light/dark theme) would make the generalization claims more convincing.

---

> ### Author Response · Authors · 2025-11-25
> **Response to reviewer**
>
> Thanks for constructive feedback!
>
> > Q. Please provide formal metrics for exploration quality.
>
> A. We focus on downstream task performance rather than exploration-specific metrics, since this is ultimately the most meaningful measure of an exploration method's value. Assessing exploration quality directly is not feasible in our setting because the GUI environments do not expose ground-truth state information. Metrics such as state coverage require knowing the total number of unique states, which we cannot access, making such measures unreliable when computed solely from exploration trajectories. Other qualitative notions like systematicity or realism are also poorly defined and difficult to measure objectively. For these reasons, we do not evaluate exploration quality directly and leave such analysis to future work.
>
> > Q. How well do AutoPlay generated tasks match natural distribution.
>
> A. We analyze this in Section 4.2 and find that the AutoPlay generated tasks do provide coverage over natural task distributions. Specifically, we present a quantitative comparison with AndroidWorld and OSWorld tasks in Fig. 3 of the main paper. We first categorize each task instruction from representative datasets into one of 13 task categories using GPT-4o. Next, we plot what percentage of total tasks in the dataset fall under a specific category for all categories and compare the task category distribution of AutoPlay with tasks from AndroidWorld and OSWorld. As shown in Fig. 3, we clearly see that AutoPlay provides good coverage over natural task distributions from these two benchmarks.
>
> > Q. Insufficient analysis of verifier accuracy.
>
> A. We perform a human evaluation to assess the reliability of our MLLM-based verifiers and find that they align well with human judgments, supporting their use in both SFT and RL. Using 200 tasks with an overall ~50% success rate, three annotators labeled each trajectory, and a demonstration was considered successful if at least two agreed. Comparing human labels with predictions from our GPT-4o and Qwen-2.5 VL Instruct verifiers, we observe ~60% accuracy and consistently high recall but low precision. This indicates that the verifiers tend to be overly optimistic, often marking tasks as successful when only partial subgoals are completed - especially for long-horizon tasks.
>
> | Verifier Model        | Accuracy | Precision | Recall |
> |-----------------------|----------|-----------|--------|
> | Qwen-2.5 VL Instruct  | 59.3     | 56.9      | 82.8   |
> | GPT-4o                | 61.8     | 57.7      | 95.9   |
>
>
> > Q. Is there a set of evaluation tasks that test OOD generalization?
>
> A. We test generalization across unseen apps, unseen env states and instructions in our experiments. To test generalization to novel apps we ensure no tasks from holdout apps are seen during training. We added these results to the revised manuscript in Tab. 3 and also present it in the response below. In addition, for results in Tab. 2 we ensure all evaluation tasks from AndroidWorld and OSWorld are deduplicated from our training dataset and use them as our OOD evaluation split. Specifically, we use a sentence embedding model to encode tasks from the benchmarks and AutoPlay generated tasks. Next, we filter out tasks that have a match greater than 0.97 similarity score. In addition, we also make sure to hold out a set of environment initial states for evaluation tasks i.e. we ensure that we do not generate any training task in the same environment initial state as any test task.
>
> > Q. Experiments for generalization to holdout apps.
>
> A. We find that agents trained with AutoPlay generalize effectively to unseen apps, achieving a 39.36% average Pass@1 success rate across five held-out applications, performance comparable to training directly on those apps. To evaluate generalization, we exclude five mobile apps (Settings, Markor, Simple Calendar Pro, Pro Expense, Broccoli–Recipe App) from the training set and test agents on them zero-shot. Given the small number of environments, we train five models, each time holding out a different app. We compare the results of zero-shot generalization to a held out environment with agents trained only on data from the held out environment. As shown in the table, agents trained on larger, more diverse app sets generalize nearly as well to new environments as agents trained exclusively on data from the test environment.
>
> | App Name              | Pass@1 | Pass@5 | Pass@1 | Pass@5 |
> |-----------------------|--------|--------|--------|--------|
> |                       | Train on Test app |        | Holdout app |        |
> | Markor                | 26.1   | 38.4   | 11.5   | 38.5   |
> | Simple Calendar Pro   | 33.6   | 70.5   | 26.8   | 47.1   |
> | Broccoli-Recipe       | 20.0   | 38.6   | 35.4   | 46.1   |
> | Pro Expense           | 47.7   | 66.6   | 51.1   | 66.6   |
> | Settings              | 74.4   | 80.0   | 72.0   | 80.0   |
> | Average               | 40.36  | 58.9   | 39.36  | 55.7   |

---

### Official Review · Reviewer_JnXJ · 2025-10-31

**Soundness:** 3
**Presentation:** 3
**Contribution:** 3
**Rating:** 6
**Confidence:** 3

**Summary:**

The paper proposes a new method, AutoPlay, for generating agent instructions given an environment. The paper shows how this method (given a strong verifier model) can be combined with rejection-sampling or RL over these synthetic instructions to improve an agent's performance, as done in prior work. This overall pipeline yields improvements on AndroidWorld and OSWorld. The key contribution is how AutoPlay conditions instruction generation on environment exploration. The paper shows how their particular method for doing this has advantages over prior work.

**Strengths:**

The proposed method is intuitively compelling. Conditioning environment exploration on previous rollouts encourages diversity. Conditioning task generation on environment exploration encourages both feasibility of generated instructions and diversity. While the general idea of synthetic instruction generation conditioned on exploration is not novel, some of the specific design decisions are different from prior work and the ablations indicate these are useful.

**Weaknesses:**

* The main novel aspect is generation of the instructions (not post-training a model given these instructions). It would have been useful to see more head-to-head comparisons and analysis with alternative recipes and methods from prior work for instruction generation, although I understand it is hard to quantify their utility without also tuning downstream models. As a practitioner, the main take-away is what aspects of the instruction generation method should be adopted. Table 2 provides some of this analysis, although perhaps additional analysis could be useful. This shouldn't necessarily block acceptance, just a suggestion.
* Relatedly, it seems most of the method's advantage comes from prompt engineering of the task guidelines. Perhaps ablations could be extended to consider which aspects related to exploration are most crucial.

Nits:

* There is some prior work on web agents that is not discussed. The recipes are similar enough to mobile and compute use agents that it is certainly relevant. For example, https://arxiv.org/abs/2410.02907 and https://arxiv.org/abs/2403.08140.

**Questions:**

* Do you plan to release the generated trajectories?

---

> ### Author Response · Authors · 2025-11-25
> **Response to reviewer**
>
> Thanks for the constructive feedback!
>
> > Q. Missing head-to-head comparisons with methods from prior works for instruction generation.
>
> A.  The No Exploration baseline is an instantiation of prior work PAE [1] and Iterative Exploration is an instantiation of prior work AgentSynth [2] with prompts tuned to UI settings, that we compare with our method in Table 2. Specifically, the No Exploration baseline instantiates PAE, which only uses pretrained knowledge of an LLM with app descriptions to propose tasks. In contrast, the Iterative Exploration baseline instantiates methods AgentSynth and Explorer methods with UI specific prompts using an iterative approach of chaining task proposer step with task execution to synthesize tasks and demonstrations. Instead of labelling these baselines as PAE or AgentSynth we label them as No Exploration and Iterative Exploration to distinguish them by the type of exploration skill used for task synthesis.
>
> > Q. Most benefits seem to come from prompt engineering in task guidelines.
>
> A. Using task guidelines is critical to ensure the generated tasks provide broader task coverage which leads to better quality datasets and large improvements in downstream tasks. However, task guidelines alone are not sufficient to achieve the best results.  As shown in Table. 2, when we compare rows 1 (No Exploration with task guidelines), 3 (AutoPlay without task guidelines) and 4 (AutoPlay) we can clearly see that environment exploration combined with task guideline prompts leads to best results. These results clearly demonstrate environment exploration or task guideline prompts alone do not lead to significant improvements. However using them in conjunction leads to best results.
>
> > Q. Missing discussion of prior works NNetNav and BAGEL.
>
> A. Thanks for pointing this out. We have added discussion comparing prior works with our method in the related work section.
>
> > Q. Do you plan to release generated trajectories?.
>
> A. Releasing generated trajectories is contingent on legal approvals from our organization. However, we do plan to release relevant prompts, data generation codebase to enable reproduction of our entire pipeline.
>
> We are happy to answer any follow-ups or concerns reviewer has. If they think our response has addressed their concerns we'd appreciate if they could acknowledge it and adjust the rating for our paper.
>
>
> [1] Proposer-agent-evaluator (PAE): Autonomous skill discovery for foundation model internet agents. Yifei Zhou, Qianlan Yang, Kaixiang Lin, Min Bai, Xiong Zhou, Yu-Xiong Wang, Sergey Levione, and Erran Li.
>
> [2] Agentsynth: Scalable task generation for generalist computer-use agents. Jingxu Xie, Dylan Xu, Xuandong Zhao, and Dawn Song.
>
> [3] Explorer: Scaling exploration-driven web trajectory synthesis for multimodal web agents. Vardaan Pahuja, Yadong Lu, Corby Rosset, Boyu Gou, Arindam Mitra, Spencer Whitehead, Yu Su, and Ahmed Hassan Awadallah.
>
> [4] Insta: Towards internet-scale training for agents, 2025. Brandon Trabucco, Gunnar Sigurdsson, Robinson Piramuthu, and Ruslan Salakhutdinov.

---

### Official Review · Reviewer_37mc · 2025-11-01

**Soundness:** 3
**Presentation:** 3
**Contribution:** 2
**Rating:** 6
**Confidence:** 3

**Summary:**

This paper introduces AutoPlay, a scalable pipeline that generates synthetic tasks for training UI agents without human annotation by first exploring interactive environments to discover states and functionalities, then using these exploration trajectories with task guideline prompts to generate diverse, feasible, and verifiable tasks. The method generates 20k tasks across 20 Android apps and 10k tasks across 13 Ubuntu applications, which are used to synthesize demonstrations via an MLLM executor and verifier. Training agents on AutoPlay-generated data improves success rates on AndroidWorld and OSWorld, with an additional gain when using reinforcement learning with verifier rewards. The approach outperforms prior methods that either use limited environment information or iterative exploration, demonstrating that exhaustive exploration combined with guided task generation produces higher-quality training data for UI agents.

**Strengths:**

1. The method is simple and practical to implement, following a straightforward two-stage pipeline of exploration. It provides an effective solution for large-scale synthetic task generation, demonstrating strong improvements for both SFT and RL.

2. The paper excels in clarity of presentation with well-structured sections and intuitive visual examples. The writing is smooth and clear, making the core contributions immediately accessible to readers.

**Weaknesses:**

1. The comparison with baselines is limited to only No Exploration and Iterative Exploration methods, missing evaluations against more sophisticated task generation approaches.
2. The paper evaluates only on AndroidWorld and OSWorld benchmarks, insufficient for demonstrating generalization across UI domains.
3. The verifier's accuracy is never validated despite being critical for filtering trajectories and providing RL rewards - no false positive/negative rates reported.
4. Experimental details are insufficiently specified - specific apps, exploration steps, task sampling strategies, and app selection criteria are missing, hindering reproducibility.

**Questions:**

1. Are there comparison results with existing task generation methods?
2. Are there evaluation results in other GUI agent benchmarks?
3. Do you validate the accuracy of the proposed verifier?
4. Is it possible to disclose more details concerning the task generation process?

---

> ### Author Response · Authors · 2025-11-25
> **Response to reviewer**
>
> Thanks for constructive feedback!
>
> > Q. Limited comparison with only No Exploration and Iterative Exploration baselines. Missing evaluation with more sophisticated task generation methods.
>
> A. The No Exploration baseline is an instantiation of prior work PAE [1] and Iterative Exploration is an instantiation of prior work AgentSynth [2] with prompts tuned to UI settings, that we compare with our method in Table 2. Specifically, the No Exploration baseline instantiates PAE, which only uses pretrained knowledge of an LLM with app descriptions to propose tasks. In contrast, the Iterative Exploration baseline instantiates methods AgentSynth and Explorer methods with UI specific prompts using an iterative approach of chaining task proposer step with task execution to synthesize tasks and demonstrations. Instead of labelling these baselines as PAE or AgentSynth we label them as No Exploration and Iterative Exploration to distinguish them by the type of exploration skill used for task synthesis.
>
> > Q. Paper only evaluates on AndroidWorld and OS-World benchmarks.
>
> A. Most prior works [1, 2, 3, 4] that study synthetic task proposal methods only demonstrate results on benchmarks from a single domain. In contrast, we demonstrate our method is applicable to two standard benchmarks from different domains (mobile and computer-use), clearly demonstrating effectiveness of our method. In addition, we show models trained on AutoPlay data generalize to unseen applications within AndroidWorld in results in table below. We leave application to additional GUI-use benchmarks as future work.
>
>
> | App Name              | Pass@1 | Pass@5 | Pass@1 | Pass@5 |
> |-----------------------|--------|--------|--------|--------|
> |                       | Train on Test app |        | Holdout app |        |
> | Markor                | 26.1   | 38.4   | 11.5   | 38.5   |
> | Simple Calendar Pro   | 33.6   | 70.5   | 26.8   | 47.1   |
> | Broccoli-Recipe       | 20.0   | 38.6   | 35.4   | 46.1   |
> | Pro Expense           | 47.7   | 66.6   | 51.1   | 66.6   |
> | Settings              | 74.4   | 80.0   | 72.0   | 80.0   |
> | Average               | 40.36  | 58.9   | 39.36  | 55.7   |
>
> > Q. Insufficient analysis of verifier accuracy.
>
> A. We conducted a human evaluation study to validate verifier effectiveness and find that our verifiers achieve high agreement with human judgement validating our choice of using MLLM as a verifier. We find the verifiers we used for SFT and RL show ~60% accuracy compared to ground truth labels and have high recall. Specifically, we sampled 200 unique tasks with ~50% execution success rate and asked 3 human annotators to label them. We label a demonstration as successful if at least 2 annotators agree on it. Next, using human labels as ground truth we compare accuracy, precision and recall for our GPT-4o and Qwen-2.5 VL Instruct model based verifiers in table below. We find both verifiers achieve ~60% accuracy and have high recall but low precision. This suggests these models are more optimistic than pessimistic in labelling a task as successful. However, we find these verifiers, even though imperfect, are still a good starting point for training models that generalize well on downstream tasks as demonstrated by performance on holdout tasks in AndroidWorld and OSWorld. Improving the verifiers will lead to better quality datasets which can be investigated as part of future work.
>
> | Verifier Model        | Accuracy | Precision | Recall |
> |-----------------------|----------|-----------|--------|
> | Qwen-2.5 VL Instruct  | 59.3     | 56.9      | 82.8   |
> | GPT-4o                | 61.8     | 57.7      | 95.9   |
>
> > Q. Details about training apps, exploration steps, task sampling strategy, and app selection criteria are missing.
>
> A. These details are already provided in Appendix C.1. Specifically, we provide hyperparameters of task generators (including exploration steps, number of tasks per guideline, number of apps, etc) for both AndroidWorld and OSWorld as part of the supplementary material of our original submission. We also provide details of how we choose the applications from AndroidWorld and OSWorld for training data collection in Section 3.1 of our main paper. We have added all of these details in the main paper as well.
>
> [1]  Proposer-agent-evaluator (PAE): Autonomous skill discovery for foundation model internet agents. Yifei Zhou, Qianlan Yang, Kaixiang Lin, Min Bai, Xiong Zhou, Yu-Xiong Wang, Sergey Levione, and Erran Li.
>
> [2]  Agentsynth: Scalable task generation for generalist computer-use agents. Jingxu Xie, Dylan Xu, Xuandong Zhao, and Dawn Song.
>
> [3] Explorer: Scaling exploration-driven web trajectory synthesis for multimodal web agents. Vardaan Pahuja, Yadong Lu, Corby Rosset, Boyu Gou, Arindam Mitra, Spencer Whitehead, Yu Su, and Ahmed Hassan Awadallah.
>
> [4]  Insta: Towards internet-scale training for agents, 2025. Brandon Trabucco, Gunnar Sigurdsson, Robinson Piramuthu, and Ruslan Salakhutdinov.

---

### Author Response · Authors · 2025-11-25
**Response to Reviewers**

We thank reviewers for their constructive feedback!

We are happy that they found our method simple, clear, intuitive and practical to implement (37mc, JnXJ, 3kvg), our proposed pipeline general and effective solution for task synthesis for UI agents (37mc, 3kvg), our writing and presentation clear (37mc, 3kvg), our design decisions distinct from prior works (JnXJ, Pkt3), our methods show significant gains across established benchmarks (37mc, 3kvg, Pkt3), our ablations thorough and explaining each design decision (3kvg). We address specific comments below and will incorporate relevant feedback.

---

### Meta-Review · Area_Chair_Uxc4 · 2026-01-06

**Summary:**

This paper proposes AutoPlay, a method that generates synthetic tasks for training agents. It generates 20k tasks across 20 Android apps and 8k successful trajectories, and 10k tasks across 13 Ubuntu apps with 3.5k successful trajectories. Experiments show improvement on success rates on AndroidWorld and OSWorld. It shows significant performance gain against existing methods.

**Reviewer Concerns:**

The main concerns are resolved:
1. Narrow benchmarking & poor generalization evidence
2. Unvalidated critical components such as GPT4O verifier.
3. Insufficient experimental rigor, such as a detailed description of implementation details.
4. Insufficient analysis.

Some concerns are still outstanding:
The practicality and scalability of autoplay.

**Reviewer Scores:**

6, 6, 6, 6

---

### Decision · Program_Chairs · 2026-01-26

Accept (Poster)